# Genetic Variation and Sex-Based Differences: Current Considerations for Anesthetic Management

**DOI:** 10.3390/cimb47030202

**Published:** 2025-03-18

**Authors:** Stephen DiMaria, Nicholas Mangano, Adam Bruzzese, Benjamin Bartula, Shruti Parikh, Ana Costa

**Affiliations:** 1Department of Anesthesiology, Renaissance School of Medicine at Stony Brook University, Stony Brook, NY 11794, USA; stefano.dimaria@stonybrookmedicine.edu (S.D.); nicholas.mangano@stonybrookmedicine.edu (N.M.); shruti.parikh@stonybrookmedicine.edu (S.P.); 2Renaissance School of Medicine, Stony Brook University, Stony Brook, NY 11794, USA; adam.bruzzese@stonybrookmedicine.edu (A.B.); benjamin.bartula@stonybrookmedicine.edu (B.B.)

**Keywords:** genetics, sex differences, genetic differences, perioperative, anesthesiology, narcotics, neuromuscular blockade, awareness, delirium, postoperative nausea and vomiting (PONV)

## Abstract

Biomedical sciences have made immense progress and numerous discoveries aimed at improving the quality of life and life expectancy in modern times. Anesthesiology is typically tailored to individual patients as its clinical effects depend on multiple factors, including a patient’s physiological and pathological states, age, environmental exposures, and genetic variations. Sex differences are also paramount for a complete understanding of the effects of specific anesthetic medications on men and women. However, women-specific research and the inclusion of women in clinical trials, specifically during child-bearing years, remain disproportionately low compared to the general population at large. This review describes and summarizes genetic variations, including sex differences, that affect responses to common anesthetic medications such as volatile anesthetics, induction agents, neuromuscular blocking drugs, opioids, and local anesthetics. It also discusses the influence of genetic variations on anesthesia outcomes, such as postoperative nausea and vomiting, allergic reactions, pain, depth of anesthesia, awareness under anesthesia and recall, and postoperative delirium.

## 1. Introduction

The practice of anesthesiology is based on a combination of several principles including physiology, pharmacology, and pharmacokinetics. Individuals with similar physical and physiologic profiles may respond differently to the same anesthetic due to the influence of multiple factors such as age, environmental exposures, and genetic variations [1]. Pharmacogenetics encompasses the broad and expanding field of studying how genetic variations affect pharmacologic activity and may help explain differences in individual responses to anesthesia [2]. Elements of interest in pharmacogenetics include metabolic enzymes, receptors, and transport channels [3].

Genetic variations range from specific gene mutations, many of which involve substitution or deletion of a single nucleotide, to broader differences including male versus female sex. Despite the seemingly obvious differences between both sexes, it remains an understudied area with a particular lag in women-specific research. This gap is not unique to anesthesiology. It was only in 1990 that the United States National Institutes of Health created guidelines for the inclusion of both sexes in clinical trials with separate analyses of trial results for each sex [4]. Studies from 2000 and 2008 have shown that women were not included in mixed-sex cardiovascular trials in a proportionate amount compared to the disease prevalence in the general population [5,6]. Globally, women remain underrepresented in clinical trials [7]. For example, in 2020, women comprised 51% of clinical trial participants but 77% of patients diagnosed with thyroid cancer worldwide [7]. However, efforts have been made to promote diversity and inclusion in clinical trials. In the United States, the US Food and Drug Administration complies with Drug Trials Snapshots annually to provide transparency regarding trial participants [8]. Furthermore, the US National Institutes of Health published the NIH-Wide Strategic Plan for Research on the Health of Women 2024–2028 to improve recruitment policies for women in biomedical research trials [9].

In recent decades, and particularly in the last 20 years, attention has been directed to investigating sex differences and additional genetic differences relevant to anesthesiology. Commonly used agents, including propofol, volatile anesthetics, neuromuscular blockers, and opioids, have shown differences in responses based on sex [10,11,12]. Understanding sex differences in enzyme activity and metabolism of medications could help explain these findings. Genetic causes for complications of anesthesia, such as postoperative nausea and vomiting (PONV), have also been investigated [13]. Several polymorphisms across multiple genes have been identified and associated with different outcomes. Of these polymorphisms, some have been linked to an increased incidence of PONV while others conversely were associated with a lower incidence of PONV [14,15,16]. The 5-hydroxytryptamine type 3 (5-HT3) receptor is the target for the widely used antiemetic ondansetron. Polymorphisms in the gene can predict a decreased response to this class of medications and may explain why some patients have minimal response to ondansetron [15]. With knowledge of genetic variants, anesthetic plans can be better tailored to prevent adverse outcomes and alternative anti-emetics can be chosen to avoid medications with a higher chance of failure. Additionally, understanding these genetic variants and the effects they exert in drug pharmacokinetics and pharmacodynamics can help prevent adverse drug reactions. Knowledge beforehand of medication dosage based on an individual’s genetic makeup can maximize the therapeutic effects and minimize adverse drug reactions, as women are more likely to be hospitalized secondary to an adverse drug reaction and have nearly two-fold greater risk than men for experiencing side effects across a plethora of drug classes [17,18]. In fact, single nucleotide polymorphisms (SNPs) can alter the structure and function of drug-metabolizing enzymes and have been shown to account for 80% of individual features in response to medications [19]. Given the potential effect genetics and sex can have on the pharmacokinetics and pharmacodynamics of medications, this review was undertaken to collect the evidence available in the literature and provide clinical context for this information.

This review describes and summarizes genetic variations, including sex differences, that affect responses to common anesthetic medications and anesthesia outcomes such as PONV, allergic reactions, pain, depth of anesthesia, awareness under anesthesia and recall, and postoperative delirium. A literature search was performed using the PubMed database to search for articles that discussed the role of sex and genetics in the field of anesthesiology. In addition, clinical correlations with regard to the findings from the literature are presented.

## 2. Anesthetic Drugs

### 2.1. Volatile Anesthetics 

#### 2.1.1. The Relationship Between Genetics and Volatile Anesthetics 

There is a wide variety of individual differences in sensitivity to volatile anesthetics and, given the largely unmetabolized nature of these agents, it is hypothesized that the mechanisms underlying those differences are genetic and involve polymorphisms of transport and target proteins [20]. In a large observational study of 150 patients having otolaryngology surgery, the patients were genotyped and indicators of their sedative response to sevoflurane, a volatile anesthetic, were recorded. Those sedative responses included BiSpectral index (BIS™ Monitoring System—Medtronic) values, end-tidal sevoflurane, and clinical scales of sedation. The researchers found nine single-nucleotide polymorphisms (SNPs) that were significantly associated with patients’ sedative and cardiovascular responses to sevoflurane. Specifically, three SNPs found in the methionine synthase reductase (MTRR) gene, which encodes methionine synthase, an enzyme involved in folate absorption, were associated with higher sensitivity to sedation by sevoflurane. Additionally, an SNP in the CABRG1 gene, which encodes the gamma-aminobutyric acid (GABA) receptor and is indirectly agonized by volatile anesthetics, was associated with increased cardiovascular sensitivity to sevoflurane. A possible mechanism of this is related to the known expression of GABA_A_ receptors in the paraventricular nucleus of the hypothalamus that normally serves to suppress sympathetic excitation [21]. Zhang et al. found that sevoflurane interrupts folate metabolism and leads to demyelination in a mouse model [22]. Furthermore, the primary anesthetic effect of inhaled anesthetics has been shown to be mediated by the α1 subunit of the GABA_A_ receptor [23].

Similarly, a prospective observational study of 500 patients undergoing abdominal surgery separated the patients into high sevoflurane-sensitivity and low sevoflurane-sensitivity groups based on the end-tidal sevoflurane to maintain BIS between 40 and 60. Both groups then underwent genome-wide association studies (GWAS) and whole exome sequencing, which identified several genes associated with sevoflurane sensitivity, including FAT2 (atypical cadherin), ADI1 (acireductone dioxygenase), NEDD4 (E3 ubiquitin protein ligase), and FOXRED2 (FAD-dependent oxidoreductase) [24]. While these genes encode proteins with known functions that may be related to sevoflurane pharmacodynamics, such as mitochondrial function and excitatory neurotransmission, their causal relationship to sevoflurane sensitivity cannot be established without laboratory models of genetic variants demonstrating gene-dependent sensitivity to sevoflurane. 

Genetic modulation in folate metabolism is also responsible for differing degrees of homocysteine elevations following nitrous oxide, a commonly used inhaled anesthetic gas (not considered a volatile agent). Nitrous oxide inhibits enzymatic processes requiring a vitamin B12 cofactor by irreversibly oxidizing cobalt atoms present in vitamin B12. One enzymatic process involves the folate cycle, in which methionine synthase demethylates 5-methyl-tetrahydrofolate to promote folate synthesis and, in the process, methylates homocysteine to produce methionine. Methylenetetrahydrofolate reductase (MTHFR) regenerates 5-methyl-tetrahydrofolate to promote the folate cycle. Inhibition of this cycle results in inadequate folate production and accumulation of homocysteine [25]. In a study of 140 patients receiving at least 2 h of 66% nitrous oxide, homozygous MTHFR polymorphisms were associated with a significantly higher elevation in homocysteine concentration versus wild-type MTHFR patients [26].

Patients with mitochondrial defects, most commonly involving nicotinamide adenine dinucleotide dehydrogenase (complex I of the electron transport chain), exhibit increased sensitivity to volatile anesthetics, and an increased propensity to develop toxicity [27]. This is primarily through the inhibition of complex I, which is already susceptible in patients with preexisting electron transport chain defects. Leigh syndrome, the most common mitochondrial disorder in children, involves the homozygous loss of function of the gene encoding mitochondrial complex proteins, resulting in seizures and hypersensitivity to volatile anesthetics [28]. Interestingly, the ketogenic diet reduces the reliance on the electron transport chain’s oxidative phosphorylation and has been used prophylactically in the prevention of seizures in these patients [29]. In a mouse model of Leigh syndrome in which the gene Ndufs4 is knocked out, the mice also exhibit exquisite sensitivity to volatile anesthetics. In a study of Ndufs4 models in which mice are fed a ketogenic diet prior to exposure to volatile anesthetics for a sham procedure, there was a significant increase in lactic acidemia and intraoperative mortality [30]. This, along with the potential for acidosis in patients with mitochondrial disorders who may have undergone prolonged preoperative fasting, necessitates caution during exposure to mitochondrial stressors (i.e., volatile anesthetics) intraoperatively. 

Of note, while there are many phenotypically distinct ryanodine receptor 1 and calcium channel subunit A1 S genetic polymorphisms, current guidelines released by the Clinical Pharmacogenetics Implementation Consortium (CPIC^®^) recommend the avoidance of triggering volatile anesthetics or succinylcholine if a patient presents with any one of the 50 identified polymorphisms [31]. Due to the poor sensitivity of genotypic diagnosis of malignant hyperthermia sensitivity, the in vitro contracture test remains the gold standard for diagnosis. 

#### 2.1.2. The Relationship Between Sex and Volatile Anesthetics 

It has been well established that female patients are less sensitive to volatile anesthetics than their male counterparts [32]. In mice models of volatile anesthesia using isoflurane, female mice had both slower induction and faster emergence than male mice. Emergence was determined using the sticker-removal test, during which an adhesive sticker is placed on the mouse’s nose, prompting immediate removal in an awake mouse. This difference was observed despite measuring identical isoflurane concentrations in the brain, suggesting that there are independent pharmacokinetic differences [33]. Interestingly, a small multicenter observational study also observed slower induction and faster emergence in women; however, they noted no changes in intraoperative electroencephalographic activity, suggesting that different pharmacogenetic phenomena influence induction, maintenance, and emergence of anesthesia [34].

A potential mechanism of this sex difference has been recently elucidated in mice, in which equal concentrations of volatile anesthetics were given to male and female mice. Administration of testosterone to the female mice rendered them more sensitive to the anesthetics [33]. These findings, along with the absence of sensitization with oophorectomy, highlight testosterone as a potential modulator of volatile anesthetic sensitivity. It is well established that the hypnotic effect of volatile anesthetics is at least in part modulated by the ventral hypothalamus, which receives hormonal modulation [35]. The finding by Wasilczuk et al. that female mice exhibit fewer active sleep-promoting neurons in this region of the hypothalamus further supports the role of testosterone in its modulation. 

### 2.2. Induction Agents 

#### 2.2.1. The Relationship Between Genetics and Induction Agents 

Induction agents are medications that are used by anesthesiologists to sedate patients undergoing anesthesia. Examples of commonly used agents include propofol, etomidate, thiopental, dexmedetomidine, and ketamine, each with distinct sedative properties. In total, 1000 patients undergoing breast cancer resection under total intravenous anesthesia with propofol were assessed in an observational study relating total propofol requirements to SNPs identified by GWAS. Two SNPs were identified that had statistically significant associations with varying effects of propofol, ROBO3-rs997989, and NALCN-rs9518419, both of which are variants in noncoding genetic sequences and possibly modulate the expression of nearby genes FEZ1 and NALCN, respectively [36]. FEZ1 is expressed primarily in glutamatergic and GABAergic neurons; however, its expression is primarily during the central nervous system (CNS) development. FEZ1 expression has been demonstrated to be “coupled” to the expression of SNARE complex proteins, the formation of which can be inhibited by propofol through its disabling interaction with the SNARE complex protein syntaxin 1A [37]. Additionally, NALCN encodes the sodium leak channel which has been observed to increase sensitivity to general anesthesia in multiple mammalian species [38]. In a different GWAS of 179 patients, SNPs in 5HT2A (encoding serotonin 2A receptor) were associated with a 20% decrease in propofol requirement and 40% decrease in onset time, while SNPs SCN9A (sodium channel), GABAA1 (GABA A-a1 receptor), GABAA2 (GABA A-a2 receptor), and CHRM2 (muscarinic acetylcholine receptor) were associated with increased cardiovascular sensitivity to propofol [39]. Similar to the discoveries of SNPs’ association with varying sensitivity to volatile anesthetics, these potentially relevant proteins require further investigation to establish a causal relationship between their polymorphism and clinically significant variations in propofol sensitivity. Ultimately, sensitivity to propofol is multi-faceted and SNPs identified should not be taken in isolation with regard to dosing of propofol in patients as further research is needed to further elucidate the genetic basis of susceptibility to propofol.

Like volatile anesthetic agents, parenteral induction agents exhibit similar degrees of mitochondrial enzyme inhibition that render patients with mitochondrial disorders vulnerable. The primary action of these medications is indirect agonism of the GABA_A_ receptor, with the exception of ketamine, an N-methyl-D-aspartic acid (NMDA) receptor antagonist. Medications such as etomidate, propofol, and thiopental inhibit complex I and significantly depress mitochondrial function [40]. Research in human subjects is sparse. However, animal models of mitochondrial dysfunction indicate increased sensitivity to most parenteral anesthetics [41]. Propofol infusion syndrome, characterized by an acute insufficiency of long-chain acylcarnitine ester transport and coexisting dysfunction of complexes I, may be more prevalent in patients with mitochondrial disorders [42]. 

Dexmedetomidine, a pure α2 adrenergic agonist, is also used as a general anesthetic in specific cases and can offer patients titratable degrees of sedation. The effect of polymorphism in the promoter region of the ADRA2A gene encoding the α2 receptor was investigated in a study of 110 patients requiring postoperative dexmedetomidine at 1.4 mcg/kg/h, wherein polymorphisms in the ADRA2A gene were identified using polymerase chain reaction and related to clinical measures of sedation. C1291G, C1291C, and G1291G polymorphisms were identified in 45.5, 43.6, and 10.9% of patients, respectively, and the C1291C and G1291G variants were significantly associated with a greater degree of sedation [43]. 

Ketamine, another parenteral drug that can be used to induce general anesthesia, is highly metabolized by hepatic enzymes CYP2B6 and CYP3A4, each of which has variability in expression and enzymatic activity [44]. There are several variants of the CYP2B6 genes, of which the CYP2B6*6 loss of function variant is the most prevalent. In a study of 50 patients undergoing continuous ketamine infusions for the management of chronic pain, those with the CYP2B6*6 genotype had significantly lower plasma clearance of ketamine and its metabolite, norketamine [45].

#### 2.2.2. The Relationship Between Sex and Induction Agents

It has long been documented that there are differences in the response to induction agents, most well established in relation to propofol, between male and female patients. In an observational study of 60 patients undergoing surgery using propofol to maintain general anesthesia, female patients demonstrated a significantly faster wakeup time compared to male patients [12]. There have also been sex differences in dose requirements to maintain general anesthesia with propofol based on BIS values, with significantly higher doses delivered in female patients than in male patients, independently controlled for known covariates such as lean body mass and age [46]. A 2012 study by Loryan et al. assessed the pharmacokinetics of propofol metabolism in males and females, measuring metabolites of propofol glucuronidation and hydroxylation by the UGT1A9 and CYP2B6 liver enzymes, respectively. Female patients were found to have significantly higher concentrations of these metabolites, as well as significantly higher expression of these liver enzymes, than male patients. Interestingly, this was unrelated to the rates of known UGT1A9 and CYP2B6 polymorphisms (e.g., CYP2B6*6), which were found to not significantly differ between male and female patients [47]. 

There is sexual dimorphism in the response of the GABA_A_ receptor to stimulation by propofol as well. In a study that involved bathing immature mouse ventromedial hypothalamic neurons with propofol, neurons from male mice more rapidly exhibited GABA_A_-driven calcium influx and more quickly returned to baseline levels compared to those from female mice. This study uniquely assessed hypothalamic dimorphism prior to developmental masculinization. Furthermore, no attenuation of the observed differences occurred when the investigators introduced testosterone in vitro [48]. Research relating the menstrual cycle to changes in response to anesthesia has been largely equivocal and limited to animal models. In one rat study, the animals were found to have statistically significant differences in time to emergence from anesthesia induced by dexmedetomidine, but not by propofol or volatile anesthetics. Furthermore, there were no changes in estrogen or progesterone concentrations that were associated with the differences observed in dexmedetomidine [49]. This demonstrates that, if menstruation confers variability in response to dexmedetomidine-induced anesthesia, the exact mechanisms remain elusive and are unlikely to be solely related to fluctuations in serum hormone concentrations.

### 2.3. Neuromuscular Blocking Drugs

#### 2.3.1. The Relationship Between Genetics and Neuromuscular Blocking Drugs

The genetic variations affecting the activity of neuromuscular blocking drugs are primarily driven by genetic variability in the expression of enzymes that are responsible for their metabolism. Butyrylcholinesterase (BChE), also called plasma cholinesterase or pseudocholinesterase, is a nonspecific plasma enzyme that is responsible for metabolizing acetylcholine and its structural analogs (e.g., benzocholine, succinylcholine), into inactive metabolites. BChE is variably expressed in individual patients and its functional activity can be deficient due to heterozygous or homozygous genetic variation. The fact that BChE levels vary in individuals in a genetically driven pattern was identified as early as 1957, when families with individuals particularly sensitive to succinylcholine were identified, and esterase levels were measured and determined to be reduced in an autosomal recessive pattern [50]. This was determined using the dibucaine-resistance test, where the ester local anesthetic dibucaine was used to inhibit BChE’s metabolism of benzocholine. If reduced inhibition was noted (<70%), an individual had abnormal or atypical BChE. Furthermore, the degree of reduced inhibition signified heterozygous (40–70%) or homozygous (<20%) inheritance pattern. The clinical consequence of expressing homozygous dibucaine-resistant BChE is an estimated paralysis time of greater than 2 h after succinylcholine administration, in contrast to the 5 min experienced by wild-type individuals [51]. More recently, up to 65 genetic variants of BChE deficiency have been identified, each with varying degrees of clinically significant alteration of activity. Nonetheless, BChE deficiency is exceedingly rare, occurring in as few as 1 in 5000 people. Therefore, genetic testing is typically reserved for patients with an individual or family history evocative of prolonged neuromuscular blockade following succinylcholine [52]. The obvious weakness of this history-based screening is its insensitivity to patients without prior personal or familial history of receiving general anesthesia, and the risk of emergence from anesthesia with persistent neuromuscular blockade presents significant problems for these patients. A recent case series suggests that the use of quantitative neuromuscular monitoring, when applied prior to the expected recovery of neuromuscular function, can detect BChE deficiency in patients early enough to allow the anesthesiologist to prevent premature emergence from anesthesia in patients with delayed metabolism of neuromuscular drugs [53].

Individual differences in the effects of nondepolarizing neuromuscular blocking drugs have also been identified, although the pathophysiology is still elusive. In a study of 230 Chinese recipients of general anesthesia, two specific SNPs were independently correlated to statistically significant differences in the duration of rocuronium-induced neuromuscular blockade [54]. Specifically, patients with NR1I2-rs2472677 experienced longer time to recovery of 25% and 90% train-of-four ratio (TOFR), and those with SLCO1A2-rs4762699 experienced shorter times to recovery of 25% and 90% TOFR. An additional study of 900 patients undergoing general anesthesia with rocuronium identified two additional SNPs, SLCO1A2-rs7967354 and SLCO1A2-rs11045995, that accounted for 41% of the variability in rocuronium dose administered to maintain 0–10% TOFR [55]. This suggests an increased role of the SLCO1A2 gene, which encodes the transport protein OATP1A2, in modulating the pharmacodynamics of rocuronium. OATP1A2 is an organic anion transporting polypeptide and is ubiquitously expressed in mammals. Recent data implicate its role in transporting β-amyloid proteins in the brain, as well as transporting bile acids in the liver and small intestine [56]. While preclinical data demonstrate the efficacy of targeting OATP1A2 in models of GI tumors, data remain limited regarding the specific function of OATP1A2 in regulating nondepolarizing neuromuscular blocking drugs. 

#### 2.3.2. The Relationship Between Sex and Neuromuscular Blocking Drugs

The primary sex-specific circumstance in which neuromuscular blockade has been demonstrably altered is in pregnancy. This occurs primarily due to the various physiologic changes in pregnancy, including increased glomerular filtration rate, total body weight, total body water, and decreased albumin levels [57]. Pharmacokinetic changes that occur in pregnancy carry clinical significance with respect to neuromuscular blockade. Pancuronium has demonstrated a 27% decreased duration of action during cesarian sections when compared to non-obstetric surgery, likely attributable to the renal clearance of pancuronium and the elevated glomerular filtration rate. Conversely, atracurium, which is not renally cleared but rather eliminated via Hofmann elimination and ester hydrolysis, has not demonstrated a change in the duration of action in pregnant patients [57].

Magnesium, which may be given to preeclamptic patients, can potentiate neuromuscular blockade due to its antagonism with postsynaptic calcium. Furthermore, in specific preeclamptic patients or those experiencing HELLP syndrome, impaired liver function results in impaired BChE production and activity correlating to elevations in aminotransferase levels, increasing the duration of action of the depolarizing neuromuscular blocking drugs [58]. Moreover, in pregnant women with a deficiency in BChE, succinylcholine may cross into fetal circulation at significant concentrations and result in neuromuscular blockade of the neonate, which is more profound if the neonate is also BChE deficient [59]. Independent of pregnancy, clinical determination of neuromuscular blockade may also vary based on a patient’s sex. A study using quantitative neuromuscular monitoring demonstrated a sex-specific discrepancy between the TOFR and clinical muscle function. In female patients, TOFR was lower before signs of paralysis (e.g., inability to raise the head, swallow, or open the eyes) manifested [60].

### 2.4. Opioids

#### 2.4.1. The Relationship Between Genetics and Opioids

In anesthesiology, opioids are useful for perioperative analgesia, sedation, pain management, reduction in nociception, and hemodynamic maintenance [61]. Commonly used opioids (fentanyl, remifentanil, hydromorphone, and morphine) agonize µ-opioid receptors in the central nervous system, including areas such as the brain stem, thalamus, and spinal cord. The three types of opioid receptors, which are all G-coupled protein receptors, include µ (MOP) (bound by morphine), delta (DOP), and kappa (KOP) receptors [62]. When bound, opioids downregulate the cAMP pathway, specifically decreasing the production of adenylate cyclase. Opioids close voltage-gated Ca^2+^ channels, which mediates the presynaptic inhibition of substance P and glutamate release. K⁺ channels are also opened when opioids bind to their receptor, allowing for postsynaptic hyperpolarization, excitability suppression in neurons, and reduction in pain signaling in the descending inhibitory pain pathways [63].

Genetic variations and sex differences have been shown to significantly influence the effects of opioids, leading to a variation in pain perception, adverse effects, and metabolism in patients. Opioid anesthetics, such as fentanyl, morphine, hydromorphone, and remifentanil, can be affected by variations in the following genes: *OPRM1*, *CYP2D6*, *UGT2B7*, *ABC1*, and *MC1R*. Primarily, fentanyl is utilized for anesthetic induction, maintenance, and postoperative pain management [61]. It is classified as a synthetic opioid and piperidine derivative with a rapid onset (1–2 min intravenously) and short duration of action (30–60 min). Through piperidine-induced oxidative dealkylation and hydroxylation, fentanyl is metabolized to norfentanyl by cytochrome P450 *CYP3A4* in the liver. Morphine, a natural opioid with a longer onset (10–30 min intravenously) and longer duration (3–4 h), is often used for preoperative sedation, intraoperative analgesia, and postoperative pain management. Primarily, the *OPRM1* gene controls μ-opioid receptor signaling in the brain; *OPRM1* genetic variants, specifically the 118 A>G variant, can alter the opioid response [64]. Kong et al. conducted a meta-analysis of 137 studies, with 17 eligible studies and 4690 patients, finding that individuals with the presence of at least one *OPRM1*-118G allele showed a reduced effect of morphine compared to individuals with the 118 A/A genotype. The *OPRM1* 118G allele was also associated with increased side effects such as vomiting after surgery. In addition to *OPRM1*, the *CYP2D6* gene is involved in metabolizing several opioids. Thus, *CYP2D6* polymorphisms can impact opioid (hydromorphone and remifentanil) potency and adverse effects through the accumulation of the original drug or its metabolites [65]. Vieira et al. employed a systematic review of 27 publications and found that *CYP2D6* polymorphisms were correlated to variation in opioid consumption between several ethnic groups [65]. Secondly, an association was detected between *OPRM1* A118G polymorphisms (rs1799971) and differences in morphine consumption between individuals. Since the *UGT2B7* gene is responsible for morphine metabolism by glucuronidation, *UGT2B7* polymorphisms have been shown to alter morphine response [66]. Bastami et al. piloted a pharmacogenetic association study that investigated the impact of *UGT2B7*, *OPRM1*, and *ABCB1* polymorphisms on morphine behavior in 40 Japanese patients undergoing abdominal hysterectomy [66]. After morphine was administered via patient-controlled analgesia (PCA), plasma concentrations of morphine and its metabolites were measured in blood samples to assess the pharmacokinetics, while pharmacodynamic effects were studied through the assessment of pain relief. Results revealed that patients with homozygosity of *UGT2B7* 802C required significantly lower doses of morphine for alleviation of pain, while the same trend was observed in homozygosity of *ABCB1* 1236T and 3435T, and of *OPRM1* 118A. More specifically, there was an enhanced correlation between *UGT2B7* T802C and morphine dosages required for pain alleviation. This study also carried out regression analysis, finding that 30% of diverse variations affecting morphine dose were linked to SNPs in *UGT2B7*, *OPRM1*, and *ABCB1* genes.

Similar to morphine, some metabolic products of fentanyl are also reportedly metabolized by glucuronate conjugation and may be affected by *UGT2B7* variations [67,68]. Muraoka et al. conducted an observational genetic association study that investigated the relationship between SNPs in the *UGT2B7* gene and individual sensitivity to fentanyl among 353 healthy Japanese patients (125 males and 228 females) aged 15–52 years scheduled for cosmetic orthognathic surgery [67]. The *UGT2B7* gene was again found to affect morphine metabolism, with the C allele of rs7439366 associated with higher levels of morphine-3-glucuronide and morphine-6-glucuronide. However, it was also found that this SNP was associated with the analgesic effects of fentanyl, indicating the C allele of the rs7439366 SNP may enhance analgesic efficacy with both fentanyl and morphine. Furthermore, there were two *UGT2B7* SNPs, rs4587017 and rs1002849, with a high likelihood of affecting the analgesic action of fentanyl. Additional studies have revealed novel genes involved in fentanyl action. Saiz-Rodríguez et al. conducted a pharmacogenetic study to explore polymorphisms altering fentanyl pharmacokinetics, pharmacodynamics, and safety profiles [69]. 35 healthy volunteers (19 men and 16 women) each received 300 μg of fentanyl (oral route) followed by genotyping for nine polymorphisms in the following genes: cytochrome P450 enzymes *CYP3A4* and *CYP3A5*, ATP-binding cassette subfamily B member 1 (*ABCB1*), opioid receptor µ1 (*OPRM1*), catechol-O-methyltransferase (*COMT*), and adrenoceptor beta 2 (*ADRB2*). This study demonstrated that *CYP3A4**22 allele carriers exhibited decreased enzyme expression and lower clearance for fentanyl, indicating slower fentanyl metabolism. Additionally, *ABCB1* 1236T/T genotypes exhibited decreased AUC, increased clearance, and shorter half-life, indicating amplified efflux of fentanyl [69]. Pharmacodynamic effects included a hypotensive effect of fentanyl across patients: *ADRB2* 523A allele correlation with reduced systolic blood pressure, and association of *OPRM1* and *COMT* gene variants with greater risk for somnolence following fentanyl administration. This study concluded that *CYP3A4*, *ABCB1*, *OPRM1*, *COMT*, and *ADRB2* polymorphisms significantly influence fentanyl’s pharmacokinetics, pharmacodynamics, and potential for adverse effects. These findings highlight the need for consideration of genetic factors in individualizing opioid therapy, minimizing adverse effects, and bolstering efficacy. Mogil et al. employed a comparative study between an animal study utilizing quantitative trait locus mapping and a human study to investigate the role of the melanocortin-1 receptor (*MC1R*) gene in mediating κ-opioid analgesia, with emphasis on sex-specific differences. In the animal study, the *MC1R* gene was found to mediate κ-opioid analgesia specifically in female mice, indicating a sex-specific genetic mechanism [70]. Besides the importance of *MC1R*, these findings indicate the critical need to consider sex-specific genetic factors in opioid use. Regarding clinical applicability, the Clinical Pharmacogenommics Implementation Consortium provides therapeutic recommendations for the use of the CYP2D6 genotype for prescribing codeine and tramadol. They recommend withholding tramadol and codeine for patients who are ultrarapid metabolizers of these drugs. However, the data regarding CYP2D6 genotype and hydrocodone, oxycodone, and methadone are weak [71].

#### 2.4.2. The Relationship Between Sex and Opioids

Considering sex differences impact on opioid potency, females generally have increased sensitivity to opioid receptor agonists compared to males [72]. Phleym et al. conducted a narrative review finding that males require 30–40% higher doses of opioid analgesics compared to females to achieve the same level of pain relief. Due to this varied sensitivity, females are at higher risk for the adverse effects of opioids such as respiratory depression. Additionally, mixed-action opioids, such as butorphanol, nalbuphine, and pentazocine, have shown increased efficacy in females than in males [73]. Packiasabapathy et al. conducted a review that attributed these sex differences to the influence of sex hormones on pharmacokinetics and pharmacodynamics, altering protein binding and metabolism of several drugs, leading to pharmacokinetic dimorphism. Furthermore, overall levels of pain perception are shown to be increased in females, where they exhibit a lower pain threshold than males [73]. Thus, these studies support that biological sex differences affect the metabolism of opioids as well as baseline pain perception.

### 2.5. Benzodiazepines

#### 2.5.1. The Relationship Between Genetics and Benzodiazepines

Benzodiazepines, such as midazolam and remimazolam, are psychoactive medications frequently indicated in anesthesiology for sedation, muscle relaxation, anxiolytic purposes, hypnosis, and anticonvulsant effects. Benzodiazepines act as gamma-aminobutyric acid (GABA-A) receptor agonists by binding to a specific site on the GABA-A receptor distinct from the GABA binding site. When bound, there is increased receptor affinity for GABA, the inhibitory neurotransmitter of the CNS, which leads to the increased influx of chloride, hyperpolarization, and succeeding suppression of neuronal excitability [74]. Due to its short elimination half-life (1–4 h), midazolam is effective for short-term sedation. Midazolam is metabolized by the cytochrome P450 system in the liver, particularly by *CYP3A4* and *CYP3A5*. Its active metabolite, hydroxymidazolam, has sedative properties. *CYP3A4* polymorphisms can significantly influence midazolam’s efficacy and clearance [75]. Elens et al. conducted an observational study that found that the *CYP3A5* *3/*3 genotype led to slower clearance rates of midazolam compared to *CYP3A5* *1/*1 or *CYP3A5* *1/*3 genotypes. Along with *CYP3A* polymorphisms, genetic variations in GABA receptor subunits have been found to influence midazolam’s sedative and amnestic properties [76]. Kosaki et al. conducted a prospective observational study of 191 patients scheduled for dental procedures with IV midazolam utilized for sedation (0.05 mg/kg). It was found that GABA A receptor β1 subunit (*GABRB1*) polymorphisms correlated with varying responses to midazolam sedation. Two SNPs in the *GABRB1* gene, rs73247636 (*p* = 0.001) and rs56278524 (*p* < 0.001), demonstrated a significant association with sedation and the occurrence of anterograde amnesia. This implies that certain *GABRB1* mutations impact the possible effects of midazolam—amnesia, sedation, and anterograde amnesia.

Remimazolam is another benzodiazepine with a short duration of action and a compound structure similar to midazolam. It is also vulnerable to genetic influences, particularly through the same *CYP3A* pathways as midazolam. Studies have indicated that *CYP3A* polymorphisms also influence remimazolam pharmacokinetics [77]. Hu et al. conducted an observational study in 62 healthy Chinese volunteers investigating how remimazolam pharmacokinetics are influenced by genetic variations in the vitamin D receptor (*VDR*), cytochrome P450 3A (*CYP3A*), and cytochrome P450 Oxidoreductase (*POR*) genes [77]. While *CYP3A* and *POR* variations were found to influence the clearance, half-life, and pharmacokinetics of remimazolam, this study found that novel *VDR* variations such as the rs4516035 allele significantly affected the elimination half-life of RF7054, the inactive carboxylic acid metabolite of remimazolam (*p* = 0.043). Additionally, association was found between the rs1544410 allele and the dose-normalized maximum observed plasma concentration (Cₘₐₓ/D) of remimazolam (*p* = 0.025). However, how the plasma concentration of benzodiazepines is related to clinical levels of sedation remains to be elucidated. Overall, *CYP3A*, *GABRB1*, *POR*, and *VDR* gene variations may all influence the pharmacokinetics and pharmacodynamics of midazolam and remimazolam. It can be concluded that pharmacogenetic testing is vital to optimizing benzodiazepine therapy, given the effect of genetic variability on benzodiazepine metabolism and response.

#### 2.5.2. The Relationship Between Sex and Benzodiazepines

Sex differences also influence the action of benzodiazepines, specifically midazolam [78]. Chen et al. utilized a meta-analysis of midazolam plasma concentration data for 118 patients from 13 studies (1996–2004) to examine the relationship between sex differences and *CYP3A* activity (key in midazolam metabolism). Following oral administration of midazolam, women were found to have 11% higher mean weight-corrected total body midazolam clearance and 28% higher clearance than men (*p* < 0.01). Although there was increased hepatic and intestinal *CYP3A* activity in women compared to men, there was a minimal difference in benzodiazepine plasma concentrations between sexes. Therefore, differences in *CYP3A* activity may or may not have significance in correlation with benzodiazepine metabolism between women and men. Two other studies investigating the pharmacokinetics and pharmacodynamic effects of remimazolam also showed no significance in the impact of sex differences on extubation time or steady-state infusion rates [79,80]. Overall, the influence of sex-based differences on the metabolism and response to benzodiazepines requires further research but should necessitate careful consideration in clinical practice.

### 2.6. Local Anesthetics

#### 2.6.1. The Relationship Between Genetics and Local Anesthetics

Local anesthetics, including lidocaine, bupivacaine, mepivacaine, and ropivacaine, mediate blockage of nerve conduction, pain alleviation, and reversible sensation loss in a specific area during minor surgeries, diagnostic procedures, and regional anesthesia (epidurals, nerve blocks). Most local anesthetics act by inhibiting sodium channels, preventing action potentials from propagating in surrounding neurons [81]. Although there is not an abundance of evidence supporting the correlation between genetic variations and local anesthetics, there are some specific genetic factors that may be related to each kind of local anesthetic. Lidocaine is commonly used due to its duration of action of 1–2 h and the possibility of extension to up to 3 h with the addition of epinephrine. Primarily, genetic variations in the melanocortin-1 receptor (*MC1R*) gene in red-haired women reduce their sensitivity to subcutaneous lidocaine and increase their sensitivity to thermal pain [82,83]. Liem et al. conducted an observation study of 60 women (30 red-haired and 30 dark-haired) to establish if the occurrence of red hair correlated with increased pain sensitivity and reduced effectiveness of topical and subcutaneous lidocaine. It was found that red-haired women presented with significantly lower thresholds for cold pain perception (22.6 °C vs. 12.6 °C; *p* = 0.004) and cold pain tolerance (6.0 °C vs. 0.0 °C; *p* = 0.001) compared to dark-haired women. Red-haired women also had slightly lower heat pain tolerance thresholds (46.3 °C vs. 47.7 °C; *p* = 0.009). There was a decreased effectiveness of subcutaneous lidocaine observed in red-haired women; red-haired women presented with a pain tolerance threshold of 11.0 mA at 2000 Hz stimulation compared to dark-haired women with a pain tolerance threshold greater than 20.0 mA at the same stimulation (*p* = 0.005). Another genetic factor affecting the response to lidocaine is the Hypoxia-Inducible Factor 1 (*HIF-1*), which was found to influence cellular resistance to lidocaine-induced toxicity [84]. Okamoto et al. conducted an experimental research study exposing lidocaine to renal cell carcinoma (RCC4-EV) cell lines that express *HIF-1* [84]. These cells exhibited enhanced resistance to lidocaine-induced cell death through suppressed mitochondrial activity, specifically the downregulation of certain components in the mitochondrial electron transport chain, indicating a change in cellular metabolism. Thus, the *MC1R* gene and *HIF-1* appear to be associated with reduced lidocaine sensitivity and increased resistance to lidocaine toxicity, respectively. Ultimately, the literature is mixed with regard to the anesthetic requirements in patients with red hair. One study in 2004 showed the minimum alveolar concentration (MAC) in volunteers to be higher in women with red hair than those with dark hair (6.2% vs. 5.2%) [82]. On the other hand, a 2012 study demonstrated no differences in MAC requirements across a broad range of surgical cases [85].

Secondly, SNPs have been associated with the administration of ropivacaine. Liu et al. employed an observation study in 100 patients to identify SNPs associated with the consumption of epidural ropivacaine during breast cancer surgery [86]. The study identified several SNPs in the *OPRM1*, *COMT*, and *CYP2D6* genes that exhibited significant association with the amount of epidural ropivacaine administered. *OPRM1* polymorphisms were linked to increased ropivacaine consumption, while *COMT* and *CYP2D6* polymorphisms were associated with varying differences in ropivacaine consumption. Thus, *OPRM1* variations may increase sensitivity to ropivacaine, while *COMT* and *CYP2D6* variably influence ropivacaine sensitivity. There has been additional evidence that supports the concept that *CYP2D6* variations may increase sensitivity to ropivacaine [87]. A study investigated a population of 256 breast cancer patients who received ropivacaine for thoracic epidural anesthesia in elective mastectomies. It was found that *CYP1A2* rs11636419 AG and GG genotypes required lower doses of epidural ropivacaine compared to patients with the AA genotype (corrected *p* = 0.024 and *p* < 0.001, respectively). Furthermore, patients with rs17861162 CG and GG genotypes required lower doses of epidural ropivacaine than those with the CC genotype (corrected *p* = 0.018 and *p* < 0.001, respectively). Thus, *CYP1A2* SNPs rs11636419 and rs17861162 may increase sensitivity to epidural ropivacaine, implying the need to personalize local anesthetic treatments for breast cancer patients. Regarding other local anesthetics such as bupivacaine or mepivacaine, there were minimal genetic associations found. However, a patient case study found that Brugada syndrome, an inherited disease related to sudden cardiac death, may be induced by bupivacaine in silent carriers of a missense mutation in the α subunit of the cardiac sodium channel (*SCN5A*) [88]. Thus, patients with *SCN5A* mutations may be vulnerable to arrhythmias and electrocardiographic abnormalities in Brugada syndrome. This has significant indications in the care of patients who develop ST-segment elevation following administration of bupivacaine, which can lead to fatality. Thus, further research is necessary to identify the pharmacogenetic side effects of bupivacaine. *SCN5A* mutations in voltage-gated sodium channels have been further shown to increase local anesthetic resistance in a small observation study investigating four family members [89]. Clendenen et al. conducted whole-exome sequencing of the four family members, finding one genetic variant in the voltage-gated sodium channel, the A572D mutation in the *SCN5A* gene, in three of the four family members. The three family members with the A572D genetic variant presented with increased resistance to lidocaine and bupivacaine while the unaffected family member responded normally to the local anesthetics [89]. Thus, *SCN5A* variations may affect the sensitivity towards bupivacaine and lidocaine.

#### 2.6.2. The Relationship Between Sex and Local Anesthetics

Sex differences show mixed effects in their association with local anesthetics. Some studies reveal increased lidocaine resistance in women [90]. Robinson et al. utilized a double-blind, randomized, placebo-controlled methodology in 44 subjects (21 females and 23 males) to test sensitivity to lidocaine between sexes [90]. They found that males in the lidocaine treatment condition rated the stimuli as less painful than the females, supporting increased male sensitivity to local anesthetics. Yet, other studies found no significant sexual differences in local anesthetic efficacy [91]. Pei et al. carried out an observational study in 60 patients aged 18–45 years to determine if there are sex differences in the minimum local analgesic concentration (MLAC) of ropivacaine for ultrasound-guided supraclavicular brachial plexus blocks [91]. This study found that the MLAC of ropivacaine did not change across the sexes. However, an additional observational study found conflicting evidence supporting greater resistance to ropivacaine in women [92]. Li et al. examined sex differences in the MLAC of ropivacaine for caudal anesthesia in 60 patients (30 men and 30 women ages 18–60 years) undergoing anorectal surgery [86]. It was found that women required a 31% higher ropivacaine MLAC for caudal anesthesia than men (*p* < 0.01). This indicates there is a greater resistance to ropivacaine in women. Yet, other factors, such as the difference in procedure, may affect sensitivity to ropivacaine across sexes. Overall, the association between sex differences and local anesthetics may depend on the type of local anesthetic used, the procedure performed, and the interplay between genetic factors, sex hormones, and individual variability.

## 3. Anesthetic Management Considerations

### 3.1. Postoperative Nausea and Vomiting

#### 3.1.1. The Relationship Between Genetics and Postoperative Nausea and Vomiting

PONV is one of the most common adverse effects of anesthesia, afflicting 30% of patients [93]. PONV affects patient experience through increased recovery time, increased cost, decreased patient satisfaction, and discharge delays. A family history of PONV as a risk factor hinted at a genetic predisposition [94]. Many of the studies investigating the association between genes and variants with PONV have used candidate genes, specifically looking at SNPs in serotonin receptors, transport proteins, cytochrome p450 enzymes, dopamine receptors, and muscarinic receptors. GWAS along with studies involving biorepositories have also been performed.

Gloor et al. genotyped 601 patients during the first 24 h after elective surgery, albeit without any antiemetic prophylaxis, and two SNPs in the *HTR3B* gene, which encode the type 3B serotonin receptor, were significantly associated with PONV occurrence (OR 1.49, *p* < 0.05) [94]. Reuffert et al. genotyped the *HTR3A* and *HTR3B* genes, which encode the subunits 5-HT_3A_ and 5-HT_3B_ respectively, in 189 German patients (95 patients experienced PONV vs. 94 control patients who did not experience PONV) undergoing elective surgery. They found the *HTR3A* c1377A>G was significantly associated with PONV (OR 2.972, *p* = 0.003) whereas the *HTR3B* variants c5+201_+202delCA (OR: 0.421, *p* = 0.001) and c6-137C>T (OR: 0.034, *p* = 0.004) were associated with a decreased risk for PONV [95]. These genetic variations are not located in the coding regions of these genes nor did they lead to amino acid exchanges, but the authors suggested they may have regulatory effects on mRNA synthesis [95]. Kim et al. in a cohort of 245 adult patients undergoing laparoscopic cholecystectomy, found patients that were a homozygous mutant for the *5HT3B* AAG deletion genotype (-100_-102AAG deletion variant) and had increased incidence of PONV within 2 h of surgery (*p* = 0.02), although no difference was noted from 2 to 24 h after surgery [15]. Furthermore, Lin et al. analyzed three SNPs of the *HTR3A* gene in a cohort of 1000 adult Taiwanese surgical patients undergoing general anesthesia [14]. Two SNPs, rs33940208 and rs10160548, were shown to be associated with postoperative nausea [14]. Additionally, the CTT haplotype was associated with postoperative nausea (OR: 2.31, *p* = 0.003), whereas the TAG haplotype was associated with a protective effect against postoperative nausea (OR: 0.28, *p* = 0.005) [14]. On the other hand, Wesmiller et al. did not find any SNPs associated with increased risk for PONV in the gene *HT3RA* gene cohort of 93 women undergoing surgery for breast cancer [96].

Multiple studies have investigated the role of muscarinic receptor polymorphisms and PONV risk. Janicki et al. performed a GWAS comparing 122 patients with severe PONV to 129 matched controls and differences in allele frequency between SNPs in the PONV and control groups [97]. They found one SNP (rs2165870) in the promoter region of the M3 muscarinic acetylcholine receptor (CHRM3) gene that was associated with PONV [97]. In addition, this GWAS study also found the voltage-gated potassium channel KCNB2 rs349358 SNP to be associated with PONV [97]. Klenke et al. confirmed both of these findings, namely the SNP in the promoter region of the M3 muscarinic acetylcholine receptor and the voltage-gated potassium channel SNP, in a prospective cohort of 454 adult patients undergoing elective surgery and a retrospective study of 474 patients undergoing elective surgery, respectively [98].

With regard to the role of dopamine receptor polymorphisms and PONV risk, Frey et al. found an association between the SNP rs1800497 DRD2 TaqIA polymorphism and patients with a history of PONV in a cohort of 306 German patients undergoing elective surgery [99]. These findings were replicated by Stegen et al. recently, in a study of German adult patients undergoing elective surgery [100]. Technically, the three SNPs, CHRM3, rs2165870, and the KCNB2 rs349358, and the recently identified dopamine D2 receptor gene SNP rs1800497, are the only three polymorphisms associated with PONV in independent studies. However, the association with the dopamine D2 receptor gene SNP rs1800497 is less clear since there are a host of studies investigating this SNP with various findings [94,101,102]. Hayase et al. found the presence of the rs3821313 or rs3755468 SNP in the *TACR1* gene, which encodes the NK-1 receptor (targeted by the antiemetic medication aprepitant) to be associated with a higher incidence of PONV [103].

Additionally, genetics influence the metabolism of opioids, which may cause PONV. A randomized controlled trial of Korean women undergoing breast surgery found that the carriers of the allele variant 118G in the human µ-opioid receptor gene (OPRM1) reported less nausea and vomiting (*p* = 0.02) [104]. In a systematic review and meta-analysis, Ren et al. found patients with the 118G variant reported less PONV than homozygous 118AA patients during the first 24 h, but not the 48 h postoperative period (OR: 1.30, *p* = 0.005) [105]. 

Ondansetron, one of the most commonly administered agents for the prevention of PONV, is metabolized by CYP2D6 enzymes [106]. The degree of CYP2D6 activity influences how patients are classified: poor metabolizers, intermediate metabolizers, normal metabolizers, and ultrarapid metabolizers [107]. The Clinical Pharmacogenetics Implementation Consortium recommends initiating ondansetron therapy with standard dosing for normal metabolizers, whereas for ultrarapid metabolizers, the Consortium recommends switching to a different agent within the class that is not metabolized by CYP2D6, such as granisetron [107]. There is insufficient data for recommendations for poor metabolizers and intermediate metabolizers [107]. One study by Candiotti et al. in 250 adult patients undergoing standard general anesthesia receiving 4 mg intravenous (IV) ondansetron found ultrarapid metabolizers had an increased incidence of postoperative vomiting, but not nausea compared to all other groups (*p* < 0.01) [108]. In 112 orthopedic trauma patients, poor metabolizers were found to have less PONV but higher pain scores after receiving 4 mg IV ondansetron [109].

Douville et al. performed a unique and detailed study by creating a polygenic risk score for the prediction of PONV in a retrospective cohort [110]. First, they performed a GWAS on a derivation cohort (5703 PONV cases, 58,820 controls), identifying 46 genetic variants, none of which were the three SNPs independently associated with PONV in the studies cited above. They then repeated the GWAS in a derivation cohort of 25,262 patients and used the associations of these genetic variants to create a polygenic score for a validation cohort consisting of 61,503 patients. Ultimately, they found when adding the polygenic risk score to traditional risk factors, there was a statistically significant, but clinically insignificant, association with PONV.

#### 3.1.2. The Relationship Between Sex and Postoperative Nausea and Vomiting

With regard to the relationship between genetics, sex, and PONV, there is a paucity of studies that focus on female-specific genetic influences on PONV. One study found the TT haplotype with two SNPs, rs3771836 and rs3755468, in the TACR1 gene was associated with reduced incidence and severity of PONV in female patients [103]. Nonetheless, the literature has clearly established that female sex is a risk factor for PONV [111]. Please refer to Table 1 for a complete list of studies summarized in this paper reflecting the relationship between sex, genetics, and PONV. 

### 3.2. Allergic Reactions Related to Anesthesia Medications

#### 3.2.1. The Relationship Between Genetics and Allergic Reactions to Anesthesia Medications

The prevalence of perioperative hypersensitivity reactions varies with estimates ranging from 1 in 20,000 in Europe, 1 in 1361 in Australia, to 1 in 385 in the United States [112,113,114]. In the United States, there was less interest in this subject, or reactions were not recognized or reported, and there are few published findings with adequate numbers of patients and new information [115]. In a study in the Chinese Han population of 89 patients who experienced anaphylaxis matched to 1339 controls, the *HLA-G*01:01* allele (OR: 2.4, *p* < 0.05) was associated with increased risk of anaphylaxis whereas *HLA-G*01:04* allele (OR: 0.3, *p* < 0.05) was protective for perioperative anaphylaxis [116]. Furthermore, studies have shown a correlation between Chinese ancestry, the HLA-B*1502 allele, and the development of Stevens–Johnson Syndrome/Toxic Epidermal Necrolysis. Specifically, the prevalence of this allele exceeds 15% in Thailand and Malaysia and is almost 10% in Taiwan, and 4% in Northern China. It has been recommended to screen for the HLA-B*1502 allele prior to initiating carbamazepine therapy in patients with East Asian or South Asian ancestry [117]. Similarly, it was recommended to test for the HLA-B*5701 in patients initiating abacavir for most patients, with the possible exception of those with ancestry from East Asia, Saudi Arabia, Ghana, and Zimbabwe [117]. Despite these findings, a perioperative risk score for anaphylaxis based on genomic markers is lacking.

Additionally, Navinés-Ferrer et al. demonstrated that in vitro mast cell degranulation by cisatracurium, morphine, and vancomycin depends on mast cell receptor MRGPRX2 expression and that this receptor is implicated in non-IgE-mediated allergic reactions [118]. Platelet-activating factor (PAF) has been shown to be an important mediator of anaphylaxis, with PAF acetylhydrolase inactivation of PAF also protecting against anaphylaxis. However, studies involving this pathway and reactions to anesthetic agents are lacking [119].

#### 3.2.2. The Relationship Between Sex and Allergic Reactions to Anesthesia Medications

The literature consistently shows females to have an increased tendency to allergic reactions [120]. In an 8-year French study of 2516 patients, adult women had three times the incidence of anaphylaxis compared to adult men with 154.9 per million procedures vs. 55.4 per million procedures, respectively [114]. In the Chinese Han population study, female sex was a risk factor for perioperative anaphylaxis as well (OR 2.8, *p* = 0.0028). Lobera et al. found that the female-to-male incidence of anaphylactic reactions under anesthesia was 3:2 in 71,063 surgical interventions [121]. Further studies are needed to elucidate the causes associated with women’s greater susceptibility to allergic reactions.

### 3.3. Pain and Analgesia

#### 3.3.1. The Relationship Between Genetics, Pain and Analgesia 

Although there have been many SNPs associated with pain sensitivity and responses to analgesic medications, the literature is conflicting [122]. Two meta-analyses help elucidate the state of the current literature regarding genetic variations, pain, and analgesia. Ren et al. performed a systematic review and meta-analysis of 23 studies (5902 patients) and found that patients with the OPRM1 118G allele variant consumed more opioids for analgesia (*p* < 0.00001); yet, they also reported higher pain scores (*p* = 0.002) compared to homozygous 118AA patients during the first 24 h postoperatively [105]. In addition, CYP3A4*IG carriers consumed fewer opioids than homozygous CYP3A4*1/*1 patients during the first 24 h postoperatively (*p* < 0.00001) [105]. The second meta-analysis of 163 studies found only two genetic variants that were associated with small differences in postoperative pain [123]. The OPRM1 A118G rs1799971 was associated with increased opioid use and postoperative pain scores (*p* < 0.00001 and *p* = 0.0004, respectively) while the COMT rs4680 SNP was associated with increased incidence of chronic (at least 3 months postoperatively) pain scores (*p* = 0.004) [123]. Finally, the Clinical Pharmacogenetics Implementation Consortium provides evidence-based recommendations for the dosing of analgesic medications based on CYP phenotype [71,124,125].

#### 3.3.2. The Relationship Between Sex, Pain and Analgesia

Various studies exist comparing sex differences in response to pain and analgesia. Although the results are mixed, the literature generally points in the direction of increased sensitivity to pain in females [126]. Furthermore, females are at increased risk for numerous chronic pain conditions [73]. Under experimental conditions, there is evidence that females have increased sensitivity to pain [127]. However, in the clinical sphere, the picture is not as clear. Miaskowski et al. reviewed 18 studies of postoperative opioid use, with 10 studies showing increased opioid use in males while the other 8 studies reported no difference between the sexes [128]. In a study of nearly 15,000 patients receiving postoperative epidural analgesia after major surgery, differences between sexes existed in numeric pain scores but were not clinically relevant [129]. Total patient-controlled epidural analgesia consumption in women was decreased compared to men, although the authors note this may be due to the increased incidence of motor blockade and vomiting amongst the female cohort. After laparoscopic sleeve gastrectomy, men reported lower pain scores in all phases of care and received more opioids during their hospitalization compared to women [130]. On the other hand, after total hip arthroplasties, men demonstrated lower odds of persistent opioid prescribing, defined as 10 or more opioid prescriptions up to 12 months post-procedure (OR: 0.90, *p* < 0.001) [131]. Niesters et al. conducted a systematic review and meta-analysis on experimental and clinical administration of opioid analgesia and found mixed results [132]. Although they found no sex differences for µ opioid analgesia across clinical studies, greater analgesic effects were found when they narrowed their analysis to patient-controlled analgesia, with even more robust findings related to morphine administration. Results were similar in the experimental studies. Another systematic review and meta-analysis found no difference in response to analgesia with ibuprofen between sexes after third molar extraction [133]. Currently, there is not enough convincing evidence to warrant sex-specific pain interventions in most situations [134].

### 3.4. Depth of Anesthesia

#### 3.4.1. The Relationship Between Genetics and Depth of Anesthesia 

The depth of anesthesia is a measure that determines the level of sedation in a patient, with general anesthesia corresponding to a patient unarousable to a painful stimulus. Smith et al. conducted a prospective study on patients with a history of awareness under general anesthesia and found that these patients are five times more likely to experience it again (RR: 5.0, 95% CI: 1.3–13.9) [135]. A genetic predisposition to awareness under anesthesia has been suggested [136]. Although a few studies found genetic associations between certain genetic markers and depth of anesthesia, nothing definitive has been established. In a study of etomidate anesthesia in 128 patients, Ma et al. found four SNPs, CYP2C9 rs1559, GABRB2 rs2561, GABRA2 rs279858, and GABRA2 rs279863, to be associated with variations in BIS values at time points ranging from 60 s to 150 s after the administration of etomidate [137]. Furthermore, the SNP UGT189, which encodes for UDP-glucuronyltransferase, an enzyme involved in the metabolism of many drugs, was found to be associated with the Extended Observer’s Assessment of Alertness and Sedation score [137]. In a novel experiment, Pavel et al. studied how the inhaled anesthetics chloroform and isoflurane activate a two-pore-domain potassium channel known as TWIK-related K+ channels (TREK-1). They then converted a non-sensitive anesthetic channel into a sensitive one in a fruit fly model they created by mutating it to be less sensitive to anesthetics [138]. Finally, one study gathered 12 patients who experienced awareness under anesthesia, compared to 12 controls, and performed whole exome sequencing [139]. They did not find any single gene variant strongly associated with intraoperative awareness or recall.

#### 3.4.2. The Relationship Between Sex and Depth of Anesthesia 

The literature on sex differences and sensitivity to anesthesia are mixed. Some studies suggest that women and men have no difference with regard to minimum alveolar concentration (MAC). An early retrospective study by Eger et al. found no difference in MAC values between men and women [140]. These findings were supported in a randomized controlled trial of 6041 patients who were randomized to BIS or end-tidal anesthetic agent concentration for prevention of awareness. The authors did not detect a difference between sexes with regard to intraoperative awareness [141]. On the other hand, Domino et al. found after analysis of 483 claims in the Closed Claims Database that women had increased odds for recall under anesthesia (OR: 3.08, 95% CI: 1.58, 6.06) [142]. A more recent study using a multicenter prospective cohort of 338 patients undergoing general anesthesia with tracheal intubation found the female sex as having an increased risk of awareness (OR: 2.7, *p* = 0.022) [143]. A recent meta-analysis of 44 studies found females had a higher incidence of awareness with postoperative recall (OR: 1.38, 95% CI: 1.09–1.75) [32]. Moreover, one study tried to isolate the effect of menstrual cycle on anesthetic requirements and found women in the follicular phase of their menstrual cycle (low progesterone) had significantly higher MAC requirements than patients in the luteal phase of their menstrual cycle (higher progesterone) with use of sevoflurane (1.55 MAC-hour vs. 1.3 MAC-hour, *p* < 0.0001) [144]. A study demonstrated the female brain in both mice and humans is more resistant to the hypnotic effects of volatile anesthetics; yet, these differences were not noticeable using cortical EEG recordings, only through analysis of subcortical sites using whole brain c-Fos mapping [33]. The method utilized to determine awareness was the ability to follow auditory cues, which was faster in females than males [28.8 min ± 8.5 min (mean ± SD) in women vs. 44.6 ± 13.8 min in men, *p* = 0.0017].

### 3.5. Intraoperative Awareness

#### 3.5.1. The Relationship Between Genetic Variations and Intraoperative Awareness

Intraoperative awareness refers to the ability of a patient to remember events that occur while they are under a therapeutic dose of anesthesia. Intraoperative awareness occurs in fewer than 0.2% of patients under general anesthesia, of which 10% to 25% experience intraoperative awareness even when the anesthetic dose is sufficient [139]. Awareness while undergoing an invasive surgical procedure is a potentially traumatizing event which should be prevented if possible. 

It has previously been reported that patients with red hair are less susceptible to anesthetic drugs and therefore require larger doses than one might predict [82,145,146,147]. More recently, Gradwohl et al. performed a cohort study that suggested that there was no significant difference in the way patients with red hair responded to anesthesia, nor was there a significant difference in rates of intraoperative awareness between red-haired patients and patients without red hair [148]. The results from Gradwohl et al.’s study contradict the earlier studies that proposed this increased anesthetic requirement in red-haired patients, suggesting that more research is necessary before a conclusion can be definitive. 

Sleigh et al. recruited patients that reported an episode of intraoperative awareness in order to perform genetic analysis [139]. They found 52 eligible patients and performed genotype analysis on 12 of them with similar phenotypes. Their study identified 29 specific genetic variations present in these 12 patients that were not present in any of their control genomes. Notably, only one of the 29 variants was present in more than one of the patients in the awareness group. Their study did not establish a causal link between any singular gene variant and intraoperative awareness; however, it does provide a foundation for more in-depth genetic research to build upon.

#### 3.5.2. Impact of Patient Sex on Intraoperative Awareness

It has been documented that women and men respond to general anesthesia differently [32,149]. Specifically, women seem to require higher doses of hypnotics and recover faster after drug delivery has stopped [149]. In a review of case reports, Ghoneim et al. found that patients who experienced intraoperative awareness were significantly more likely to be female [150]. Lennertz et al. used the isolated forearm technique to determine patients’ responses to commands while under anesthesia [143]. They found that approximately 13% of their female subjects displayed a response, compared with 6% of the male subjects [143].

Wasilczuk et al. recently tested the hypothesis that this difference is mediated by sex hormones in the body. As mentioned above, in animal models, they found that female mice had lower sensitivity to volatile anesthetics than male mice [33]. Wasilczuk et al. concluded through additional experiments that in female mice, there is lower c-Fos expression in the portions of the hypothalamus responsible for control of sleep when compared to male mice. The authors suggest that this difference in hypothalamic activity may be responsible for the increased incidence of intraoperative awareness in human female patients as well. 

### 3.6. Postoperative Delirium

#### 3.6.1. Impact of Genetic Variation on Postoperative Delirium

Postoperative delirium (POD) is a relatively common complication of anesthetic administration [151]. The occurrence of POD increases the duration of hospitalization, morbidity, and mortality [152]. There are several studies within the literature that have examined the relationship between various genetic variants and the incidence of POD. Kazmierski et al. performed a prospective case–control study to determine whether genetic variations in serotonin receptor 2a (5HT2A) or the 2B/3A subunits of the NMDA receptor (GRIN2B and GRIN3A, respectively) might be associated with differences in the incidence of POD [153]. They reported that there was a significant increase in the rate of postoperative delirium in those patients expressing the AG genotype of the 3A subunit of the NMDA receptor when compared to the GG genotype. However, the strength of these findings is limited due to a relatively small sample size. 

Mahanna-Gabrielli et al. investigated the role of a SNP expressed in the melatonin receptor 1B gene in POD [152]. The authors found that there was a significant increase in the rate of POD in individuals expressing the risk genotype when compared to those who did not. This study is also limited by a small sample size and only patients undergoing cardiac surgery were included. Of note, patients undergoing cardiac surgery are particularly susceptible to POD [151]. Heinrich et al. performed a GWAS and a candidate gene-associated study to test the hypothesis that neurotransmitter imbalance is responsible for the development of POD, specifically the role of cholinergic genes [151]. Ultimately, they identified three SNPs in the genes coding for M2 and M4 muscarinic receptors that were significant for the development of POD. Two other GWAS were performed, where the first study found an SNP in SLC9A4 (encodes a Na/H pump in the stomach) and the second found two SNPs: FHIT (Fragile histidine triad) and SUGCT (Succinyl-CoA-glucotarate-CoA-transferase) associated with delirium from patients in the RIPHeart cohort [154,155]. In a case–control study, Terrelonge et al. explored SNPs in the genes FKBP5, KIBRA, KLOTHO, MTNR1B, and SIRT1 [156]. These genes were selected due to their known association with either cognition or delirium. Of these five genes, SNPs in KIBRA, FKBP5, and MTNR1B were associated with an altered likelihood of developing POD based on the specific genotype present. Interestingly, the MTNR1B SNP is the same as that studied earlier by Mahanna-Gabrielli et al. [152], confirming the findings of that publication in a larger sample. Additionally, Terrelonge et al. investigated patients who underwent elective orthopedic surgery as opposed to cardiac surgery. Overall, studies involving the genetics of POD are still scarce, and more research is needed to adequately determine which genes, if any, have implications for the development of POD. Please refer to Figure 1 for a summary of the genetic variations that affect the metabolism of anesthetic agents.

#### 3.6.2. Impact of Patient Sex on Postoperative Delirium 

Literature concerning sex as a risk factor for POD appears to be conflicting. Some previous reviews have established male sex as a risk factor for POD, while others have failed to make the same determination [157,158,159,160,161]. Moreover, Mevorach et al. draw attention to possible evidence of publication bias and confounding of statistical analyses in some studies. Overall, evidence for male sex as a risk for developing POD seems to be weak [158]. Figure 2 depicts a summary of the response to anesthetic drugs on the basis of sex. Table 2 collectively summarizes the studies investigating the relationship between genetics, sex, and anesthetic agents. 

## 4. Discussion

The implications of genetic variations and sex differences affect the activity of nearly every anesthetic medication. Theoretically, by identifying genotypes associated with altered responses to a certain medication, the anesthetic plan can be tailored to either avoid that medication or dose it in a way to improve the outcome. Interestingly, there are several genetic polymorphisms that demonstrate clinical significance across different classes of medications used in anesthesia care. Perhaps most obvious are the polymorphisms in the CYP genes, which predictably alter the pharmacodynamics of drugs metabolized by their corresponding enzyme. Differences in the duration of action of several anesthetic medications that belong to different classes (e.g., ketamine, ropivacaine, midazolam, fentanyl, ondansetron) have been associated with polymorphisms in CYP genes [45,69,74,87,109]. Additionally, OPRM1 polymorphisms are associated with different pharmacodynamic responses to local anesthetics (e.g., ropivacaine) and opioid analgesics (e.g., fentanyl) [67,86]. Similarly, the MC1R polymorphisms identified in individuals with red hair have been associated with different sensitivity to local anesthetics as well as opioids [70,82]. Polymorphisms in the CHRM2 gene were associated with different cardiovascular responses to induction doses of propofol, and independently with differences in postoperative delirium rates [39,151]. However, another opportunity for future research relates to the specific mechanisms by which these polymorphisms are associated with their distinct clinical effects across classes of anesthetic medications.

Sex differences pertinent to females include increased rates of awareness during general anesthesia, shorter duration of time for emergence and waking up, and decreased sensitivity to induction agents. These factors suggest female patients may require higher doses of anesthesia medications. Different dosage guidelines for male and female patients may be necessary to guide clinicians. However, further investigation would be required to determine the optimal dose for each sex. Outcomes from these studies would be relevant to another growing topic in anesthesiology: target-controlled infusions (TCIs). The role of TCIs in anesthesia is to optimize the delivery of infusions by computing patient-specific doses to obtain a predetermined effect using pharmacokinetic and pharmacodynamic models that account for factors including age, sex, height, and weight [162,163]. When developing these models, it is crucial to account for sex-related differences to increase the accuracy of infusion doses. 

While there is a long list of genetic determinants that have been identified to be pertinent to anesthetic outcomes, the clinical applicability of this information is unclear. Genetic analysis is costly and not all health insurance plans cover genetic testing [164,165]. While the cost per test has decreased from several thousands of US dollars to now under US $250, access to genetic testing remains a limiting factor globally [164]. For lower-income countries, genetic testing may not be readily available, and when it is available, the cost is prohibitively expensive [164]. Additional barriers to access include race, poverty, and underinsurance. Racial minorities make up a small fraction of results in genomic databases, with a majority of genetic test results consisting of Caucasian patients from high-income countries [166,167,168]. Minority groups are also less likely to have knowledge about or access to genetic testing [169]. Having a balanced database is necessary to accurately determine genetically based risks for all racial and ethnic groups [168]. Due to cost and limited access to this technology, genetic testing is far from becoming routine medical evaluation. However, the long-term solution does not consist of performing genetic tests on all patients requiring anesthesia. Future investigation on the cost-effectiveness of genetic testing in preventing adverse anesthesia outcomes would help identify situations where genetic testing would be of value. A potential role for genetics in anesthesia would be post-anesthetic testing for patients with severe or unexplained complications during anesthesia. In the setting of a known, identified issue, a genetic study may provide insight into a potential cause. 

## 5. Conclusions

Consideration of genetic factors in individualizing drug therapy to optimize efficacy and minimize adverse effects is a key aspect of patient safety and best practices in medicine. Anesthesiology administration is tailored to an individual based on multiple parameters such as physiology, pathology, age, weight, comorbidities, response to individual drugs, specific procedures, or surgery, among others. Pharmacogenetic discoveries are contributing to a greater understanding of the interaction of drugs with specific individuals or groups. Further research on how genetic variations and sex differences affect the action and side effects profile of anesthetic medications and common anesthesia complications is essential for the creation of more specific patient-tailored anesthetic management plans. The practice of a patient-tailored individually based medical management plan with data input from pharmacogenetics and large data group analyses, in conjunction with the latest technological advances, appears to be approaching in the near future. A system of patient data documentation based on genetic testing and data analyses in large patient data algorithms could be a potential way to optimize the latest advancements in medicine tailored to a specific patient. However, there remain significant limitations to the clinical applicability of genetically tailored anesthesia care. Primarily, there is a strong need for future studies that randomize individuals with and without specific genetic polymorphisms to different anesthetic regimens. Nonetheless, establishing a causal relationship between genetic differences and clinical observations remains difficult to achieve in humans. More animal studies with direct gene-editing to assess the clinical influences of experimental polymorphisms may help elucidate causality. Additionally, polymorphisms may be unique to specific ethnic groups, and thus more broadly sampled genetic surveys are needed to better characterize the variety of genetic factors influencing response to anesthesia. Given the need for more robust genetic information obtained for understudied groups, such as ethnic minorities, and the high cost of routine genetic testing, its utility for changing anesthetic care decisions remains limited.

## Figures and Tables

**Figure 1 cimb-47-00202-f001:**
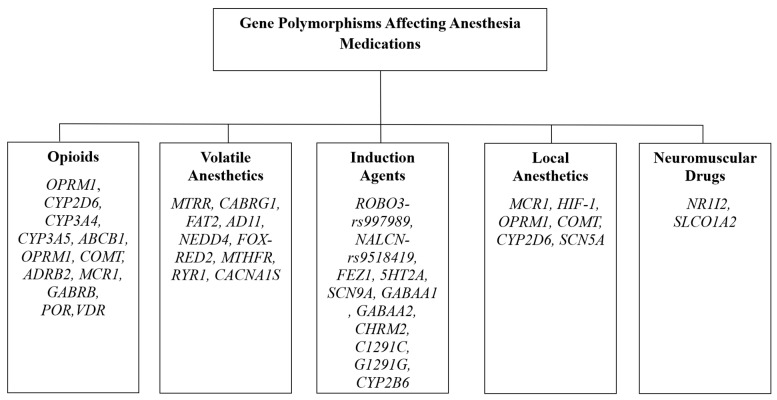
Summary of genetic polymorphisms affecting anesthetic medications.

**Figure 2 cimb-47-00202-f002:**
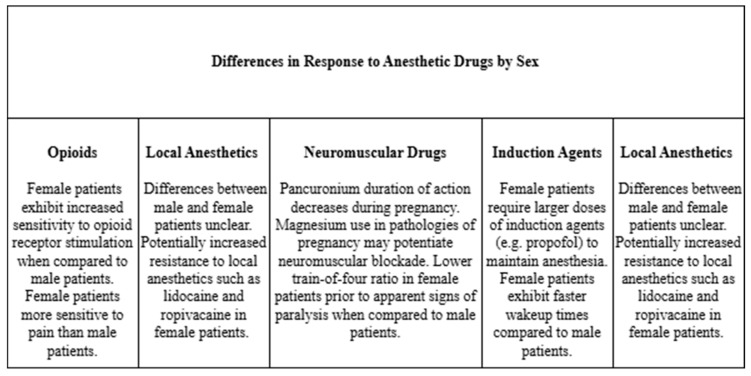
Summary of response to anesthetic drugs on the basis of sex.

**Table 1 cimb-47-00202-t001:** Studies investigating the relationship between genetics, sex, and PONV.

Study	Objective	Patient Population	Findings
Gloor et al. [94]	Identify genetic risk factors of PONV and use these to identify at-risk patients.	Total of 601 adult patients below ASA IV classification	Two SNPs in the type 3B serotonin receptor gene were associated with PONV.
Reuffert et al. [95]	Identify an association between genetic variants of the serotonin receptor subunits 3A and 3B and PONV.	Adult German patient population, 95 of whom have PONV after general anesthesia and 94 control patients.	*HTR3A* c1377A>G was associated with PONV whereas the *HTR3B* variants c5+201_+202delCA (OR: 0.421, *p* = 0.001) and c6-137C>T were associated with a decreased risk for PONV.
Kim et al. [15]	Investigate whether two deletion polymorphisms in the serotonin 3B receptor gene affect efficacy of ondansetron in preventing PONV.	Total of 245 adult patients undergoing laparoscopic cholecystectomy	Homozygous mutants for the *5HT3B* AAG deletion genotype (-100_-102AAG deletion variant) had increased incidence of PONV within 2 h of surgery (*p* = 0.02), although not at 2–24 h after surgery.
Joy Lin et al. [14]	Identify an association between SNPs in the serotonin 3A receptor gene and postoperative nausea.	Total of 369 adult Taiwanese patients	Two SNPs were associated with increased postoperative nausea. One haplotype showed an increased risk while one haplotype showed a protective effect.
Wesmiller et al. [96]	Identify genetic risk factors associated with PONV after surgery in breast cancer patients.	Total of 90 adult women	Alleles in genes for catchol-O-methyltransferase, dopamine receptors, and tryptophan were associated with decreased PONV.
Janicki et al. [97]	Conduct a genome-wide association study to identify novel loci for genes predisposing to PONV.	Total of 122 patients with severe PONV matched to 129 controls	One SNP in the M3 muscarinic receptor was associated with PONV.
Klenke et al. [98,101]	The primary objective was to investigate the relationship between genetic factors and the Apfel score with PONV risk. The second objective was to determine whether PONV prophylaxis with dexamethasone and acustiumlation or both decrease PONV risk for patients with genetic risk factors for PONV.	Total of 454 adult patients undergoing elective surgery	A polymorphism in the M3 muscarinic receptor is an independent risk factor for PONV and combined prophylaxis with dexamethasone and acustimulation reduced PONV rate.
Frey et al. [99]	Investigate the association of a polymorphism in the dopamine receptor with PONV in a high-risk cohort	Total of 306 German patients undergoing elective strabismus repair with etomidate/alfentanil/mivacurium induction and sevoflurane maintenance	The TaqIA A2 allele in the dopamine receptor gene is significantly associated with a history of PONV.
Stegen et al. [100]	Two objectives: (1) Create PONV prediction score which includes SNPs in the M3 muscarinic receptor gene (*CHRM3*, same SNP as Klenke et al. [98,101], and the potassium voltage-gated channel subfamily B member 2 (*KCNB2*) gene; (2) investigate association with five additional SNPs with PONV.	Total of 838 adult German patients	The *CHRM3* and *KCNB2 SNPs* were unable to be used to create a PONV prediction score. SNPs in the dopamine receptor were found to be associated with PONV.
Hayase et al. [103]	Identify if SNPs in the neurokinin-1 receptor (TACR1 gene) are associated with sex differences in PONV.	Total of 200 adult surgical patients	One SNP in the NK1 gene was associated with sex differences in PONV.
Lee et al. [104]	Identify if an SNP in the mu-opioid receptor is associated with PONV risk	Total of 416 Korean women undergoing breast surgery	PONV rates differ based on opioid receptor polymorphism with no difference in pain scores.
Ren et al. [105]	Systematic review and meta-analysis on studies investigating an association between genetic polymorphism and clinical outcomes of opioid analgesics.	Total of 23 studies (5902 patients)	Patients with a gene variant in the opioid receptor experienced less PONV during the first 24 h but not at the 48 h period.
Candiotti et al. [108]	Determine if patients who were ultrarapid metabolizers and given ondansetron had a greater rate of PONV.	Total of 250 adult female patients	Ultrarapid metabolizers had increased incidence of postoperative vomiting but not nausea.
Wesmiller et al. [109]	Investigate association of CYP2DC genotypes with PONV.	Total of 112 adult trauma patients	Poor metabolizers experienced less PONV but higher pain scores after receiving 4mg IV ondansetron.
Douville et al. [110]	There were multiple objectives to this study as it consisted of multiple stages (1): perform a genome-wide association study to identify genetic risk factors for PONV; (2): derive a polygenic risk for PONV in a derivation cohort; (3): use this polygenic risk score combined with traditional risk factors for PONV in a validation cohort; (4): compare genetic contributions to PONV with the literature.	Total of 61,503 adult surgical patients	The use of a polygenic risk score did not clinically improve PONV prediction when compared to traditional risk factors.

**Table 2 cimb-47-00202-t002:** Studies investigating the relationship between genetics, sex, and anesthetics.

Topic	Summary of Gene Mutations	Summary of Sex Differences
Volatile anesthetics	- MTRR gene polymorphisms were associated with higher sensitivity to sedation by sevoflurane [21]; - CABRG1 gene polymorphisms were associated with increased cardiovascular sensitivity to sevoflurane [21]; - FAT2, ADI1, NEDD4, and FOXRED2 gene polymorphisms were associated with differences in sevoflurane sensitivity [24]; - Homozygous MTHFR polymorphisms were associated with significantly higher homocysteine concentration elevations [26]; - Patients with defective NADH dehydrogenase exhibit increased sensitivity to volatile anesthetics and an increased propensity to develop toxicity [27]; - There are many RYR1 and CACNA1S genetic polymorphisms; however, CPIC^®^ recommends the avoidance of triggering volatile anesthetics or succinylcholine in the presence of any one of the 50 identified polymorphisms [31].	- Female mice had both slower induction and faster emergence than male mice using isoflurane [33]; - Women had slower induction and faster emergence; however, they had no changes in intraoperative electroencephalographic activity [33]; - Testosterone administration to female mice rendered them more sensitive to the anesthetics [33]; - Female mice exhibit fewer active sleep-promoting neurons in this region of the hypothalamus [35].
Induction agents	- SNPs in the ROBO3-rs997989 and NALCN-rs9518419 genes were shown to modulate the expression of FEZ1 and NALCN [36]. - FEZ1 expression can be inhibited by propofol through its disabling interaction with the SNARE complex protein syntaxin 1A [37] - NALCN expression has been shown to increase sensitivity to general anesthesia in mammals [38]; - SNPs in the 5HT2A gene were associated with a 20% decrease in propofol requirement and 40% decrease in onset time [39]; - SNPs in the SCN9A, GABAA1, GABAA2, and CHRM2 genes were associated with increased cardiovascular sensitivity to propofol [39]; - Animal models of mitochondrial dysfunction indicate increased sensitivity to most parenteral anesthetics [41]; - Propofol infusion syndrome may be more prevalent in patients with mitochondrial disorders [42]; - C1291C and G1291G polymorphisms were significantly associated with a greater degree of sedation by dexmedetomidine [43]; - CYP2B6*6 loss of function variant of the CYP2B6 gene was associated with significantly lower plasma clearance of ketamine and its metabolite, norketamine [45].	- Females demonstrated a significantly faster wakeup time from propofol compared to male patients [12]; - Significantly higher propofol doses were required in female patients than in male patients to achieve similar depths of anesthesia [46]; - Female patients were found to have significantly higher expression of the UGT1A9 and CYP2B6 liver enzymes than male patients [47]; - Neurons in male mice more rapidly exhibited GABA_A_-driven calcium influx following propofol administration and more quickly returned to baseline levels compared to those from female mice [48]; - There were significant differences in time to emergence from dexmedetomidine based on stage in the menstrual cycle in rats [49]; - There were no associations between the time to emergence from dexmedetomidine and estrogen or progesterone concentrations [49].
Neuromuscular blocking drugs	- BChE levels vary in an autosomal recessive pattern [50]; - Homozygous dibucaine-resistant BChE is related to an estimated paralysis time of greater than 2 h after succinylcholine administration, in contrast to the 5 min experienced by wild-type individuals [51]; - The NR1I2-rs2472677 SNP was associated with longer times to recovery of 25% and 90% TOFR [55]; - The SLCO1A2-rs4762699 SNP was associated with shorter times to recovery of 25% and 90% TOFR [55]; - The SLCO1A2-rs7967354 and SLCO1A2-rs11045995 SNPs accounted for 41% of variability in rocuronium dose administered to maintain 0–10% TOFR [55];	- Pancuronium has demonstrated a 27% decreased duration of action during cesarian section versus during non-obstetric surgery, likely attributable to the renal clearance of pancuronium and the elevated GFR that is a physiologic change during pregnancy [57]; - Atracurium has not demonstrated a change in duration of action in pregnant patients [57]; - Preeclamptic patients or those experiencing HELLP syndrome have impaired BChE production and activity, increasing the duration of action of the DNMBDs [58]; - In pregnant women with a deficiency in BChE, succinylcholine may cross into fetal circulation and result in neuromuscular blockade of the neonate, which is more profound if the neonate is also BChE deficient [59]; - In female patients, TOFR was lower before signs of paralysis (e.g., loss of head raise, swallowing, eye-opening) manifested [60].
Opioids	- *OPRM1* 118 > G polymorphism affects morphine efficacy and increases side effects like vomiting [64]; - *CYP2D6* polymorphisms affect the metabolism of opioids such as morphine, fentanyl, hydromorphone, and remifentanil [65]; - *UGT2B7* polymorphisms affect metabolism and dose requirements of morphine and fentanyl [66,67,68]; - *CYP3A4, CYP3A5, ABCB1, OPRM1, COMT, ADRB2* are related to fentanyl metabolism and response [69]: ○ *CYP3A4* *22 allele and *ABCB1* 1236T/T genotype were associated with slower fentanyl metabolism; ○ *ADRB2* 523A allele correlated with tendency toward reduced systolic blood pressure; ○ *OPRM1* and *COMT* gene variants were associated with higher risk factors for the development of somnolence following fentanyl administration. - *MC1R* gene variants may alter analgesic efficacy of pentazocine in red-haired individuals [70]; - Genetic polymorphisms in CYP3A4 can significantly influence the efficacy and clearance of midazolam [75]; - *GABRB1* variations influence midazolam sedative and amnesic effects [76] - Remimazolam pharmacokinetics are affected by *CYP3A*, *GABRB1*, *POR*, and *VDR* gene variations [77].	- Females generally have increased sensitivity to opioids and require lower doses for similar pain relief compared to males [72]: ○ Females are more sensitive to opioid-related adverse effects like respiratory depression. - Mixed-action opioids, such as butorphanol, nalbuphine, and pentazocine, have shown increased efficacy in females than males [73]; - Pain perception is generally higher in females, with a lower pain threshold than males [133]; - Women have reduced clearance and prolonged half-life of midazolam compared to males [78]; - The clearance of remimazolam is reported to be 11% higher in females than males, but further studies are needed [79,80].
Local anesthetics	- *MC1R* gene variations reduce sensitivity to lidocaine in red-haired individuals and increase pain perception [82]. - *HIF-1* gene is linked to resistance to lidocaine toxicity in renal cells [84]; - *OPRM1*, *COMT*, and *CYP2D6* polymorphisms affect the consumption of ropivacaine [86]: ○ *OPRM1* polymorphisms were linked to increased ropivacaine consumption. - *CYP2D6* variations increase sensitivity to ropivacaine [87]; - Brugada syndrome may be induced by bupivacaine in silent carriers of *SCN5A* missense mutations [88]; - *SCN5A* mutations can cause increased resistance to bupivacaine and lidocaine [89].	- Studies show mixed results regarding sex differences in local anesthetic efficacy; - Males have been found to be more sensitive to lidocaine compared to females [90]; - There are no sex differences in the minimum local analgesic concentration (MLAC) of ropivacaine for ultrasound-guided supraclavicular brachial plexus blocks [91]; - Women required a 31% higher ropivacaine MLAC for caudal anesthesia than men undergoing anorectal surgery (*p* < 0.01) [92].
Allergic reactions	- The *HLA-G*01:01* allele (OR: 2.4, *p* < 0.05) was associated with increased risk of anaphylaxis whereas *HLA-G*01:04* allele (OR: 0.3, *p* < 0.05) was protective for perioperative anaphylaxis [116]; - In vitro, mast cell degranulation by cisatracurium, morphine, and vancomycin depends on mast cell receptor MRGPRX2 expression and that this receptor is implicated in non-IgE-mediated allergic reactions [118]; - Platelet-activating factor (PAF) has been shown to be an important mediator of anaphylaxis, but studies with anesthetic agents are lacking [119].	- In general, the literature consistently shows females to have an increased tendency to allergic reactions [120].
Pain and analgesia	- The OPRM1 118G allele variant was associated with increased opioid requirements and higher pain scores during the first 24 h postoperatively [105]; - The OPRM1 A118G rs1799971 was associated with increased opioid use and postoperative pain scores while the COMT rs4680 SNP was associated with increased incidence of chronic (at least 3 months postoperatively) pain scores (*p* = 0.004) [123].	- The literature is conflicting. Females are at increased risk for numerous chronic pain conditions, although there is not enough convincing evidence to warrant sex-specific pain interventions in most situations [73,134].
Depth of anesthesia	- In summary, a couple of studies found genetic associations between certain genetic markers and depth of anesthesia, but nothing definitive has been established; - Those with a history of awareness under general anesthesia and found those with this prior history are five times more likely to experience it again (RR: 5.0, 95% CI: 1.3–13.9) [135]; - Four SNPs to be associated with BiSpectral Index values [137]; - Chloroform and isoflurane were found to activate a two-pore-domain potassium channel known as TWIK-related K+ channels (TREK-1) [138].	- Results have been mixed but lean towards females having higher requirements for anesthesia; - A meta-analysis of 44 studies found females had a higher incidence of awareness with postoperative recall (OR: 1.38, 95% CI:1.09–1.75) [32].
Intraoperative awareness	- Melanocortin 1 receptor mutation—possible association with decreased susceptibility to anesthetic drugs [82,145,146]; - Twenty-nine other gene variants identified only in patients reporting intraoperative awareness [139].	- Women require higher doses of hypnotics and recover faster than men after drug delivery has ended [149]; - Patients experiencing intraoperative awareness are more likely to be female [143,150].
Post-op delirium	- Reported SNPs associated with altered likelihood of POD: - GRIN3A SNP rs3739722 [153]; - CHRM2 SNPs rs8191992 and rs6962027 [151]; - CHRM4 SNP rs2067482 [151]; - MTNR1B SNP rs10830963 [156]; - KIBRA SNP rs17070145 [156]; - FKBP5 SNP rs1360780 [156].	- Weak evidence suggesting male sex increases likelihood of POD [157,160,161]; - Possible publication bias and confounding of statistics [158].

## Data Availability

The original data presented in the study are openly available in PubMed at https://pubmed.ncbi.nlm.nih.gov/.

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
