# Peer review of "Genetic Variation and Sex-Based Differences: Current Considerations for Anesthetic Management"

_cimb, 2025, doi:10.3390/cimb47030202_

Round 1

Reviewer 1 Report

Comments and Suggestions for Authors

The manuscript covers a broad range of topics but could benefit from a more focused structure. Could the authors consider organizing sections more explicitly into pharmacokinetics, pharmacodynamics, and clinical implications?

The title suggests a focus on sex-based differences and genetic variation. However, the balance between these two topics varies across sections. Could the authors ensure consistency in discussing both aspects equally?

Terms like "sex differences" and "gender differences" are used interchangeably. Since "sex" refers to biological differences and "gender" to sociocultural identity, should the authors standardize the terminology?

While each section presents relevant studies, there is limited discussion on how findings from different anesthetic classes relate to each other. Would the authors consider adding a summary that synthesizes key patterns across drug classes?

The introduction outlines the importance of genetic and sex-based differences in anesthetic responses but does not provide a clear research gap. Could the authors better articulate the novelty and need for this review?

The discussion on female underrepresentation in clinical trials is important. However, could the authors add references discussing recent improvements in gender inclusion policies?

The review discusses polymorphisms in GABA and folate metabolism genes affecting sevoflurane response. Could the authors provide more mechanistic insights or animal model evidence supporting these associations?

Testosterone is proposed as a modulator of anesthetic sensitivity. Could the authors elaborate on whether similar effects are seen with other hormones, such as estrogen or progesterone?

The review describes SNPs affecting propofol metabolism but does not mention any implications for clinical dosing. Should anesthesiologists consider different induction doses based on genetic profiling?

The section mentions up to 65 BChE variants affecting succinylcholine metabolism. Could the authors discuss the current clinical recommendations for genetic testing in preoperative screening?

CYP2D6 polymorphisms are discussed extensively regarding opioid metabolism. Could the authors clarify whether current guidelines recommend genetic screening before opioid prescription?

The section suggests CYP3A and VDR polymorphisms influence benzodiazepine metabolism. How do these findings align with clinical variability observed in sedation depth and recovery?

The review lists various SNPs associated with PONV risk. Has any polygenic risk scoring been validated to predict PONV risk in clinical settings?

The discussion on HLA alleles and anaphylaxis risk is insightful. Could the authors provide more information on how genetic testing might be integrated into perioperative allergy risk assessment?

The review highlights sex-based differences in opioid sensitivity and pain perception. Could the authors discuss whether sex differences in opioid response are independent of genetic variations?

The review briefly mentions SNPs in CYP2C9 and GABA receptors associated with BIS monitoring differences. Could the authors clarify whether these SNPs influence intraoperative awareness risk?

Previous studies suggested MC1R variants increase anesthetic requirements in red-haired patients, but recent studies challenge this. Could the authors discuss whether the MC1R-anesthesia hypothesis is still clinically relevant?

The section mentions possible genetic predispositions but lacks strong supporting evidence. Are there any GWAS studies linking specific genes to delirium risk?

While the review is comprehensive, it lacks a discussion of limitations in genetic research for anesthetic pharmacology. Could the authors acknowledge issues like small sample sizes, ethnic variability, and replication challenges in pharmacogenetic studies?

The manuscript concludes without a strong call for future studies. Could the authors suggest specific research gaps, such as personalized anesthesia dosing based on pharmacogenetic testing?

Comments on the Quality of English Language

It lacks readability. This is probably due to the lack of visual materials (Figures).

Author Response

 For review article

1. Summary

Thank you very much for reviewing our manuscript. Please find our responses below and the corresponding revisions in the re-submitted manuscript.

Point-by-Point Comment to Reviewer 1

Comment 1:

Could the authors consider organizing sections more explicitly into pharmacokinetics, pharmacodynamics, and clinical implications?

Thank you for pointing this out. While that is a valid point, we feel it will further add to the numerous sections of this paper and may hinder readability. We have attempted to describe more explicitly the pharmacokinetics and pharmacodynamics, as well as clinical implications, under each section.

Comment 2:

The title suggests a focus on sex-based differences and genetic variation. However, the balance between these two topics varies across sections. Could the authors ensure consistency in discussing both aspects equally?

Thank you for your comment. This discrepancy is currently due to the difference in available research. For example, there has been extensive research into the genetic variation regarding metabolism of drugs associated with postoperative nausea and vomiting that the reader would benefit from understanding. However, the sex-based differences regarding postoperative nausea and vomiting have been well established so we did not feel it necessary to review the literature in this section.

Comment 3:

Terms like "sex differences" and "gender differences" are used interchangeably. Since "sex" refers to biological differences and "gender" to sociocultural identity, should the authors standardize the terminology?

Thank you for this important point. We have adopted the reviewer’s suggestion and edited all mentions of “gender” to “sex” as the term “sex” is more appropriate in the context of this review.

For example, Line 748: “Various studies exist comparing sex differences in response to pain and analgesia.”

Line 756: “In a study of nearly 15,000 patients receiving postoperative epidural analgesia after major surgery, differences between sexes existed in numeric pain scores but were not clinically relevant.

Line 804: “The authors did not detect a difference between sexes with regard to intraoperative awareness.”

Comment 4:

While each section presents relevant studies, there is limited discussion on how findings from different anesthetic classes relate to each other. Would the authors consider adding a summary that synthesizes key patterns across drug classes?

Thank you for your comment. We have integrated this suggestion in the discussion (lines 907-959).

Comment 5:

The introduction outlines the importance of genetic and sex-based differences in anesthetic responses but does not provide a clear research gap. Could the authors better articulate the novelty and need for this review?

Thank you for your feedback. We have incorporated this feedback in our introduction.

Revised introduction (lines 68-79): Additionally, understanding these genetic variants and the effects they exert in drug pharmacokinetics and pharmacodynamics can help prevent adverse drug reactions. Knowledge beforehand of medication dosage based on an individuals’ genetic makeup can maximize the therapeutic effects and minimize adverse drug reactions, as women are more likely to be hospitalized secondary to an adverse drug reaction and have nearly two-fold greater risk than men for experiencing side effects across a plethora of drug classes. In fact, single nucleotide polymorphisms (SNPs) can alter the structure and function of drug metabolizing enzymes and have been shown to account for 80% of individual features in response to medication. Given the potential effect genetics and sex can have on pharmacokinetics and pharmacodynamics of medications, this review was undertaken to collect the evidence available in the literature and provide clinical context for this information.

Comment 6:

The discussion on female underrepresentation in clinical trials is important. However, could the authors add references discussing recent improvements in gender inclusion policies?

The authors appreciate this feedback and have incorporated it into the revised manuscript. Lines 42-52: “Studies from 2000 and 2008 have shown that women were not included in mixed-sex cardiovascular trials in a proportionate amount compared to the disease prevalence in the general population [5,6].  Globally, women remain underrepresented in clinical trials [7]. For example, in 2020 women comprised 51% of clinical trial participants but 77% of patients diagnosed with thyroid cancer worldwide [8]. However, efforts have been made to promote diversity and inclusion in clinical trials. In the United States, the US Food and Drug Administration complies Drug Trials Snapshots annually to provide transparency regarding trial participants [9]. Furthermore, the US National Institutes of Health published the NIH-Wide Strategic Plan for Research on the Health of Women 2024-2028 to improve recruitment policies for women into biomedical research trials [10].”

Comment 7:

The review discusses polymorphisms in GABA and folate metabolism genes affecting sevoflurane response. Could the authors provide more mechanistic insights or animal model evidence supporting these associations?

Yes, studies showing mechanistic insights were included in lines 100-108: “Additionally, a SNP in the CABRG1 gene, which encodes the gamma-aminobutyric acid (GABA) receptor and is indirectly agonized by volatile anesthetics, was associated with increased cardiovascular sensitivity to sevoflurane. A possible mechanism of this is related to the known expression of GABAA receptors in the paraventricular nucleus of the hypothalamus that normally serve to suppress sympathetic excitation [22]. Zhang et al. found in a mouse model that sevoflurane interrupts folate metabolism and leads to demyelination [23]. Furthermore, the primary anesthetic effect of inhaled anesthetics has been shown to be mediated by the α1 subunit of the GABAA receptor [24].”

Comment 8:

Testosterone is proposed as a modulator of anesthetic sensitivity. Could the authors elaborate on whether similar effects are seen with other hormones, such as estrogen or progesterone?

Yes, there are two studies included which look at the effects of these hormones (lines 247-261). One study conducted in an animal model found no differences in progesterone or estrogen concentrations that reflected differences in dexmedetomidine emergence. In addition, we discuss another study (lines 811-815) that found patients with low progesterone had higher MAC requirements than patients with high progesterone as determined based on the phase of patients’ menstrual cycle.

Comment 9:

The review describes SNPs affecting propofol metabolism but does not mention any implications for clinical dosing. Should anesthesiologists consider different induction doses based on genetic profiling?

Ultimately there is a lack of research to warrant recommendations for different induction doses based on SNPs identified. This perspective was included in the manuscript (lines 201-204): “. Ultimately, sensitivity to propofol is multi-faceted and SNPs identified should not be taken in isolation with regards to dosing of propofol in patients as further research is needed to further elucidate the genetic basis of susceptibility to propofol.”

Comment 10:

The section mentions up to 65 BChE variants affecting succinylcholine metabolism. Could the authors discuss the current clinical recommendations for genetic testing in preoperative screening?

Thank you for your comment. The recommendation is included in lines 283-285 which states genetic testing in preoperative screening is only recommended if there is an individual or family history of prolonged paralysis after succinylcholine administration, given the rarity of pseudocholinesterase deficiency.

Comment 11:

CYP2D6 polymorphisms are discussed extensively regarding opioid metabolism. Could the authors clarify whether current guidelines recommend genetic screening before opioid prescription?

Thank you for your feedback. Yes, we have included the recommendations. Lines 424-429: “Regarding clinical applicability, the Clinical Pharmacogenommics Implementation Consortium provides therapeutic recommendations for the use of the CYP2D6 genotype for prescribing codeine and tramadol. They recommend withholding tramadol and codeine for patients who are ultrarapid metabolizers of these drugs. However, the data regarding CYP2D6 genotype and hydrocodone, oxycodone, and methadone are weak [73].”

Comment 12:

The section suggests CYP3A and VDR polymorphisms influence benzodiazepine metabolism. How do these findings align with clinical variability observed in sedation depth and recovery?

Thank you for your suggestion. The studies looking at these polymorphisms have investigated plasma concentrations of benzodiazepines but there is currently no literature providing a clinical correlation between plasma concentrations and clinical level of sedation. This was clarified in lines 485-487: “However, how the plasma concentration of benzodiazepines is related to clinical levels of sedation remains to be elucidated.”

Comment 13:

The review lists various SNPs associated with PONV risk. Has any polygenic risk scoring been validated to predict PONV risk in clinical settings?

Yes, in lines 677-685 we describe a polygenic score study that found the use of a polygenic score to not enhance PONV risk calculation with the addition of traditional PONV risk scoring.

Comment 14:

The discussion on HLA alleles and anaphylaxis risk is insightful. Could the authors provide more information on how genetic testing might be integrated into perioperative allergy risk assessment?

Thank you for your comment. There are recommendations to test for HLA-B1502 for those of Chinese ancestry in patients starting carbamazepine and abacavir. The study with the Chinese Han participants did find one allele associated with perioperative anaphylaxis although there are currently no studies that have looked into a genetic scoring system for perioperative allergy risk assessment. This information was added in lines 701-710: “Furthermore, studies have shown a correlation between Chinese ancestry, the HLA-B*1502 allele and the development of Stevens Johnson Syndrome/Toxic Epidermal Necrolysis. Specifically, the prevalence of this allele exceeds 15% in Thailand and Malaysia, and is almost 10% in Taiwan, and 4% in Northern China. It has been recommended to screen for the HLA-B*1502 allele prior to initiating carbamazepine therapy in patients with East Asian or South Asian ancestry [122]. Similarly, it was recommended to test for the HLA-B*5701 in patients initiating abacavir for most patients, with the possible exception of those with ancestry from East Asia, Saudi Arabia, Ghana, and Zimbabwe [122] . Despite these findings, a perioperative risk score for anaphylaxis based on genomic markers is lacking.”

Comment 15:

The review highlights sex-based differences in opioid sensitivity and pain perception. Could the authors discuss whether sex differences in opioid response are independent of genetic variations?

Unfortunately there aren’t any studies available that isolate the response to opioids independently of genetic variations in the context of pain perception. We clarify the studies in lines 746-770, specifically in lines 764-770: “Although they found no sex differences for µ opioid analgesia across clinical studies, greater analgesic effects were found when they narrowed their analysis to patient-controlled analgesia, with even more robust findings related to morphine administration. Results were similar in the experimental studies. Another systematic review and meta-analysis found no difference in response to analgesia with ibuprofen between sexes after third molar extraction [76]. Currently, there is not enough convincing evidence to warrant sex-specific pain interventions in most situations [138].“

Comment 16:

The review briefly mentions SNPs in CYP2C9 and GABA receptors associated with BIS monitoring differences. Could the authors clarify whether these SNPs influence intraoperative awareness risk?

Yes, SNP UGT189 was associated with the Extended Observer’s Assessment of Alertness and Sedation Score. This was included in lines 783-786: “Furthermore, the SNP UGT189, which encodes for UDP-glucuronyltransferase, an enzyme involved in the metabolism of many drugs, was found to be associated with the Extended Observer’s Assessment of Alertness and Sedation score [141].”

Comment 17:

Previous studies suggested MC1R variants increase anesthetic requirements in red-haired patients, but recent studies challenge this. Could the authors discuss whether the MC1R-anesthesia hypothesis is still clinically relevant?

This area is still up for debate with no clear answer. We summarized the studies for and against this point in lines 538-543: “Ultimately, the literature is mixed with regards to the anesthetic requirements in patients with red hair. One study in 2004 showed the minimum alveolar concentration (MAC) in volunteers to be higher in women with red hair than those with dark hair (6.2% vs. 5.2%)[89] . On the other hand, a 2012 study demonstrated no differences in MAC requirements across a broad range of surgical cases [90].”

Comment 18:

The section mentions possible genetic predispositions but lacks strong supporting evidence. Are there any GWAS studies linking specific genes to delirium risk?

Thank you for your feedback. We have included the studies related to this topic. The only one GWAS study which identified three SNPs in the genes coding for M2 and M4 muscarinic receptors that were significant for development of POD. Two other GWAS studies were included in lines 878-886: “Heinrich et al. performed a GWAS and a candidate gene associated study to test the hypothesis that neurotransmitter imbalance is responsible for development of POD, specifically the role of cholinergic genes [157]. Ultimately, they identified three SNPs in the genes coding for M2 and M4 muscarinic receptors that were significant for development of POD. Two other GWAS studies were performed, where the first study found a SNP in SLC9A4 (encodes a Na/H pump in the stomach) and the second found two SNPs: FHIT (Fragile histidine triad) and SUGCT (Succinyl-CoA-glucotarate-CoA-transferase) associated with delirium from patients in the RIPHeart cohort [160,161].”

Comment 19:

While the review is comprehensive, it lacks a discussion of limitations in genetic research for anesthetic pharmacology. Could the authors acknowledge issues like small sample sizes, ethnic variability, and replication challenges in pharmacogenetic studies?

The authors appreciate this feedback and have revised our manuscript to reflect this suggestion. Lines 971-982: “However, there remain significant limitations to the clinical applicability of genetically tailored anesthesia care. Primarily, there is a strong need for future studies that randomize individuals with and without specific genetic polymorphisms to different anesthetic regimens. Nonetheless, establishing a causal relationship between genetic differences and clinical observations remains difficult to achieve in humans. More animal studies with direct gene-editing to assess for the clinical influences of experimental polymorphisms may help elucidate causality. Additionally, polymorphisms may be unique to specific ethnic groups and thus more broadly sampled genetic surveys are needed to better characterize the variety of genetic factors influencing response to anesthesia. Given the need for more robust genetic information obtained for understudied groups, such as ethnic minorities, and the high cost of routine genetic testing, its utility for changing anesthetic care decisions remains limited.”

Comment 20:

The manuscript concludes without a strong call for future studies. Could the authors suggest specific research gaps, such as personalized anesthesia dosing based on pharmacogenetic testing?

Upon the reviewers’ feedback, we have revised the conclusion to include a strong call for future studies (lines 966-982).

4. Response to Comment on the Quality of English Language

Point 1:

Response 1: It lacks readability. This is probably due to the lack of visual materials (Figures).

Thank you for your comment. We have included two tables that provide visual information and help summarize key points from each section.

Reviewer 2 Report

Comments and Suggestions for Authors

This is a long and comprehehensive review of various factors which influence the response to a variety of anaesthetic agents in the human subject .The paper concentrates on sex differnces and genetics on these different responses.It emphaasises the relative lack of information between the sesxe with less information on women

Author Response

Thank you very much for your comment and insight into our manuscript, it is greatly appreciated. 

Reviewer 3 Report

Comments and Suggestions for Authors

Comments

The review manuscript with entitled “Genetic Variation and Sex-Based Differences: Current Considerations for Anesthetic Management”. It is interesting. However.

  1. The article should have a methodology section that introduces data, information, and literature sources.
  2. The main content of each section should be represented with diagrams to facilitate readers' understanding.
  3. The format of the references is confusing, and it does not meet the requirements of the journal.

Author Response

1. Summary

Thank you very much for taking the time to review this manuscript. Please find the detailed responses below and the corresponding revisions/corrections in the re-submitted files.

The review manuscript with entitled “Genetic Variation and Sex-Based Differences: Current Considerations for Anesthetic Management”. It is interesting. However.

Comment 1:

  1. The article should have a methodology section that introduces data, information, and literature sources.

Thank you for your comment. We have incorporated the reviewer’s feedback and revised the introduction. Lines 83-96. “A literature search was performed using the PubMed database to search for articles that discussed the role of sex and genetics in the field of anesthesiology. In addition, clinical correlations with regards to the findings from the literature are presented.”

Comment 2:

  1. The main content of each section should be represented with diagrams to facilitate readers' understanding.

Thank you for this suggestion. We have included two tables that provide a visual input and summary of the studies relevant for each of the sections discussed.

Comment 3:

  1. The format of the references is confusing, and it does not meet the requirements of the journal.

Thank you for your comment. We have revised the format for the references.

Reviewer 4 Report

Comments and Suggestions for Authors

The authors have provided a comprehensive review, but sometimes interpret study results as they “need”, or insufficiently. This distorts the essence of the primary scientific literature.

e.g. in paragraph 2.4.2. Averbuch et al. carried out a meta-analysis of 314 subjects (195 female and 119 male subjects) included in the ibuprofen treatment arm of 7 doubleblind post–third-molar extraction dental pain studies [69]. The study found that females experienced greater postoperative pain than males to a statistically significant degree (P =.006). Thus, these studies support that biological sex differences affect the metabolism of opioids as well as baseline pain perception.

However, the authors write in their results and conclusions that:

Moreover, the mean pain intensity and pain relief scores over time for the female and male treatment groups were not noticeably different at any time point after drug administration, with no imputation for missing values. Our results demonstrated no sex effect on the analgesic response to ibuprofen.

Arch Intern Med. 2000;160:3424-3428.

A careful review of the entire literature with an exact assignment of the primary results is necessary.

Apart from a good recommendation for a balanced database, some more precise and detailed recommendations (i.e. a usable clinical conclusion) in the discussion would significantly enhance this work.

Author Response

For review article

Summary

Thank you very much for your comments and insights into our manuscript. We have adopted your suggestions and revised our manuscript.

Comments

Comment 1:

The authors have provided a comprehensive review, but sometimes interpret study results as they “need”, or insufficiently. This distorts the essence of the primary scientific literature.

e.g. in paragraph 2.4.2. Averbuch et al. carried out a meta-analysis of 314 subjects (195 female and 119 male subjects) included in the ibuprofen treatment arm of 7 double-blind post–third-molar extraction dental pain studies [69]. The study found that females experienced greater postoperative pain than males to a statistically significant degree (P =.006). Thus, these studies support that biological sex differences affect the metabolism of opioids as well as baseline pain perception.

However, the authors write in their results and conclusions that:

Moreover, the mean pain intensity and pain relief scores over time for the female and male treatment groups were not noticeably different at any time point after drug administration, with no imputation for missing values. Our results demonstrated no sex effect on the analgesic response to ibuprofen.

Arch Intern Med. 2000;160:3424-3428.

A careful review of the entire literature with an exact assignment of the primary results is necessary.

Response:

Thank you for your feedback. We have revised the manuscript to better reflect the reviewer’s suggestions. Lines 764-770: “Although they found no sex differences for µ opioid analgesia across clinical studies, greater analgesic effects were found when they narrowed their analysis to patient-controlled analgesia, with even more robust findings related to morphine administration. Results were similar in the experimental studies. Another systematic review and meta-analysis found no difference in response to analgesia with ibuprofen between sexes after third molar extraction [76]. Currently, there is not enough convincing evidence to warrant sex-specific pain interventions in most situations [138].”

Comment 2:

Apart from a good recommendation for a balanced database, some more precise and detailed recommendations (i.e. a usable clinical conclusion) in the discussion would significantly enhance this work.

Thank you for your comment. We have incorporated your suggestions and expanded the discussion and conclusion. Lines 965-981: “The practice of a patient-tailored individually based medical management plan with data input from pharmacogenetics and large data group analyses, in conjunction with the latest technological advances, appears to be approaching in the near future. A system of patient data documentation based on genetic testing and data analyses in large patient data algorithms could be a potential way to optimize the latest advancements in medicine tailored to a specific patient. However, there remain significant limitations to the clinical applicability of genetically tailored anesthesia care. Primarily, there is a strong need for future studies that randomize individuals with and without specific genetic polymorphisms to different anesthetic regimens. Nonetheless, establishing a causal relationship between genetic differences and clinical observations remains difficult to achieve in humans. More animal studies with direct gene-editing to assess the clinical influences of experimental polymorphisms may help elucidate causality. Additionally, polymorphisms may be unique to specific ethnic groups and thus more broadly sampled genetic surveys are needed to better characterize the variety of genetic factors influencing response to anesthesia. Given the need for more robust genetic information obtained for understudied groups, such as ethnic minorities, and the high cost of routine genetic testing, its utility for changing anesthetic care decisions remains limited.”

Round 2

Reviewer 4 Report

Comments and Suggestions for Authors

---